# Microglia promote glioblastoma via mTOR-mediated immunosuppression of the tumour microenvironment

Anaelle A Dumas[1], Nicola Pomella[1], Gabriel Rosser[1], Loredana Guglielmi[1], Claire Vinel[1], Thomas O Millner[1], Jeremy Rees[2], Natasha Aley[3], Denise Sheer[1] (ID), Jun Wei[4], Anantha Marisetty[4], Amy B Heimberger[4], Robert L Bowman[5], Sebastian Brandner[2] (ID), Johanna A Joyce[6] & Silvia Marino[1,*] (ID)

## Abstract

Tumour-associated microglia/macrophages (TAM) are the most numerous non-neoplastic populations in the tumour microenvironment in glioblastoma multiforme (GBM), the most common malignant brain tumour in adulthood. The mTOR pathway, an important regulator of cell survival/proliferation, is upregulated in GBM, but little is known about the potential role of this pathway in TAM. Here, we show that GBM-initiating cells induce mTOR signalling in the microglia but not bone marrow-derived macrophages in both *in vitro* and *in vivo* GBM mouse models. mTOR-dependent regulation of STAT3 and NF-κB activity promotes an immunosuppressive microglial phenotype. This hinders effector T-cell infiltration, proliferation and immune reactivity, thereby contributing to tumour immune evasion and promoting tumour growth in mouse models. The translational value of our results is demonstrated in whole transcriptome datasets of human GBM and in a novel *in vitro* model, whereby expanded-potential stem cells (EPSC)-derived microglia-like cells are conditioned by syngeneic patient-derived GBM-initiating cells. These results raise the possibility that microglia could be the primary target of mTOR inhibition, rather than the intrinsic tumour cells in GBM.

**Keywords** glioblastoma; microglia; mTOR; T cells; TAM
**Subject Categories** Cancer; Immunology
**The EMBO Journal (2020) 39: e103790**

## Introduction

No effective therapy currently exists for glioblastoma (GBM), which is the most common primary brain tumour and one of the most aggressive types of cancers. Challenges in tackling these tumours are manifold, including their inter- and intratumour heterogeneity, the limited accessibility of systemically administered drugs, their infiltrative growth pattern and the complexity of the microenvironment in which they are embedded (Aldape *et al*, 2019).

The contribution of the tumour microenvironment (TME), which is shaped by the communication between tumour cells and non-malignant cells, is undisputed in GBM pathogenesis (Quail & Joyce, 2017). Particular emphasis has been placed on immune infiltrates, including tumour-associated microglia/macrophages (TAM), which are the most numerous infiltrating cell population in GBM (Szulzewsky *et al*, 2015; Chen *et al*, 2017; Darmanis *et al*, 2017; Roesch *et al*, 2018). These cells engage in a bidirectional interaction with tumour cells to promote several aspects of glioma development, including proliferation, angiogenesis, immune evasion and therapeutic resistance (Hambardzumyan *et al*, 2016; Chen & Hambardzumyan, 2018). TAM in GBM are pro-tumourigenic, with increased accumulation in high-grade gliomas that correlates with poor prognosis (Komohara *et al*, 2008; Hambardzumyan *et al*, 2016; Sorensen *et al*, 2018). Moreover, TAM produce low levels of pro-inflammatory cytokines and lack key molecular mechanisms necessary for T-cell stimulation, suggesting a suppression of T-cell activation capacity in GBM (Hussain *et al*, 2007; Quail & Joyce, 2017). TAM can be classified into tumour-associated microglia (TAM-MG), endogenous to central nervous system (CNS) tissue and tumour-associated macrophages (TAM-BMDM), originating from bone marrow-derived monocytes that infiltrate the tumour from the periphery (Muller *et al*, 2015; Bowman *et al*, 2016; Haage *et al*, 2019). The functional contribution of TAM to GBM pathogenesis is well documented; however, it is unclear how each of these two ontogenetically distinct populations differentially contribute to the GBM phenotype.

Efforts are being invested in therapeutically depleting immune cells from the TME as well as altering cytotoxic potential with

1 Blizard Institute, Barts and The London School of Medicine and Dentistry, Queen Mary University London, London, UK
2 National Hospital for Neurology and Neurosurgery, University College London Hospitals NHS Foundation Trust, London, UK
3 Division of Neuropathology, Department of Neurodegenerative Disease, UCL Queen Square Institute of Neurology, London, UK
4 Department of Neurosurgery, The University of Texas MD Anderson Cancer Center, Houston, TX, USA
5 Human Oncology and Pathogenesis Program, Memorial Sloan Kettering Cancer Center, New York, NY, USA
6 Department of Oncology, Ludwig Institute for Cancer Research, University of Lausanne, Lausanne, Switzerland
*Corresponding author. Tel:+44 2078 822585; E-mail: s.marino@qmul.ac.uk

immunomodulation (Seoane, 2016). Targeting chemokines and their receptors such as the CCR2/CCL2 axes has been explored as a therapeutic strategy to inhibit infiltration of TAM (Ruffell & Coussens, 2015; Vakilian *et al*, 2017). For the re-education of TAM immune activity, inhibition of colony-stimulating factor 1 receptor (CSF1R) has shown promising result in preclinical GBM models by blocking tumour growth and progression (Pyonteck *et al*, 2013; Yan *et al*, 2017). However, acquired resistance and tumour relapse emerge following long-term exposure to these therapies (Quail *et al*, 2016). To design successful re-education strategies targeting TAM, a better characterisation of their signalling mechanisms is essential.

The mTOR pathway has been extensively studied in the context of cell growth, proliferation and survival in many cancers, including GBM (Li *et al*, 2016; Jhanwar-Uniyal *et al*, 2019). The central component of the pathway, the mTOR protein kinase, forms the catalytic subunit of the protein complexes known as mTOR complex 1 (mTORC1) and mTOR complex 2 (mTORC2), which regulate different branches of the mTOR network (Shimobayashi & Hall, 2014; Yuan & Guan, 2016). mTORC1 signalling integrates inputs from inflammatory and growth factors as well as amino acids, energy status, oxygen levels and cellular stress pathways. The two major substrates of mTORC1 are p70 ribosomal protein S6 Kinase 1 (p70S6K1) and the eukaryotic translation initiation factor (eIF)-binding protein 1 (4EBP1; LoRusso, 2016). These signalling molecules impact on cell growth and metabolism, in part by increasing the biosynthesis of the cellular translational apparatus (Thoreen *et al*, 2012). The small GTPase Ras homolog enriched in brain (Rheb) is the only known direct activator of mTORC1. Conversely, the signalling pathways that lead to mTORC2 activation are not characterised in such detail. mTORC2 is known to regulate cell cycle entry, cell survival and actin cytoskeleton polarisation through its most common downstream substrates: AKT, SGK and PKC (Yang *et al*, 2013). Despite their biochemical and functional differences, crosstalk has been reported between the two complexes, which contributes to the modulation of their activity (Xie & Proud, 2014). In GBM, altered mTORC1 signalling activity correlates with increased tumour grade and is associated with poor prognosis (Duzgun *et al*, 2016). Consequently, mTOR kinase inhibitors targeting both mTORC1 and mTORC2 are considered promising anti-cancer therapies and are being tested in clinical trials, in combination with radiation and chemotherapy (Zhao *et al*, 2017; Mecca *et al*, 2018).

In the last decade, extensive work has been carried out to characterise mTOR-dependent signalling in innate immune cells and its role in regulating the expression of inflammatory factors, antigen presentation, phagocytic activity, cell migration and proliferation (Weichhart *et al*, 2008; Jones and Pearce, 2017). mTOR signalling is known to regulate the balance between pro- and anti-inflammatory responses and may be responsible for the dysregulated inflammatory response in TAM, which display a shift towards anti-inflammatory activity. Interestingly, increased mTOR phosphorylation at Ser-2448 is present in nearly 40% of TAM in human GBM (Lisi *et al*, 2019); however, the functional impact of this mTOR deregulation and its molecular mechanism have never been characterised.

Here, we have used GBM orthotopic allografts in genetically engineered mice in which mTORC1 signalling has been silenced in TAM-MG, as well as human expanded-potential stem cells (EPSC)-derived microglial-like cells and matched GBM cells to study the role of the mTOR pathway in TAM-MG in the GBM microenvironment.

# Results

## mTOR signalling is upregulated in TAM-MG but not TAM-BMDM in GBM mouse models

To determine whether mTOR signalling was deregulated in TAM, we assessed the activity of the pathway in GBM mouse models that recapitulate the genetic signatures of human GBM: GL261 allograft model, Ntv-a;PDGFB + Shp53 (Bowman *et al*, 2016), $Pten^{-/-}$; $p53^{-/-}$ (Jacques *et al*, 2010), $Pten^{-/-}$; $p53^{-/-}$; $Idh1^{R132H}$ and PDGFB genetic model (Zhang *et al*, 2019). Tumours were stained for ionised calcium binding adaptor molecule 1 (Iba1, a marker of TAM) and for phosphorylated S6 (p-S6; S240/244, a downstream marker of mTORC1 activity). Co-expression of these markers, as defined by the fraction of Iba1$^+$ cells expressing p-S6, was quantified. Three regions—non-tumour brain tissue, tumour edge and tumour core—were defined and analysed separately, to account for potentially different functional properties of TAM within the tumour as compared to the surrounding brain, as previously reported (Darmanis *et al*, 2017; Fig 1A). Tumour core regions were defined as highly cellular areas composed almost entirely of tumour cells (cells with marked pleomorphism and nuclear atypia —increased nuclear size and hyperchromasia—as well as mitotic activity on the haematoxylin and eosin (H&E) staining). The tumour edge refers to the infiltration zone, while areas without tumour infiltration were defined as non-tumour brain tissue (Fig 1A). As expected, Iba1$^+$ cells showed a ramified morphology with long, thin cellular processes and small cell bodies in non-tumour areas, while they displayed shorter and fewer processes, with rounder cell bodies at the tumour edge. In the tumour core, they acquired an amoeboid shape, without branched processes and large round cell bodies (Fig 1B). Quantification showed that p-S6$^+$ Iba1$^+$ cells were extremely low in the non-tumour tissue (ranging from 1% ($\pm$ 0.2 SEM) in the GL261 model to 13.9% ($\pm$ 7.4 SEM) in the PDGFB model), whereas co-expression was more frequent in the tumour core (between 50% and 83%, with the PDGFB model at 83% ($\pm$ 9.9 SEM) closely followed by the GL261 model at 71% ($\pm$ 8 SEM); Fig 1B). In the tumour edge regions, all models except Ntv-aPDGFB + Shp53 showed significant difference in the fraction of Iba1$^+$ cells co-expressing p-S6 as compared to non-tumour tissue (between 31% ($\pm$ 4 SEM) the GL261 model and 72% ($\pm$ 14 SEM) in the $Pten^{-/-}$; $p53^{-/-}$ model; Fig 1B). We conclude that increased mTOR activity was consistently observed in TAM across tumour models, and it was predominantly independent of intratumour location.

Taking advantage of transcriptomic data from TAM-MG and TAM-BMDM isolated from GL261 tumours (Bowman *et al*, 2016), deregulation of the mTOR pathway was further investigated in these two cell populations. 4,907 and 5,089 deregulated genes were identified in the comparison between healthy microglia and TAM-MG, and between blood monocytes and TAM-BMDM, respectively. Amongst these deregulated genes, the majority were unique to TAM-MG and TAM-BMDM (3,405 and 3,687, respectively), in keeping with the different ontogeny of these cells (Fig 1C). The most

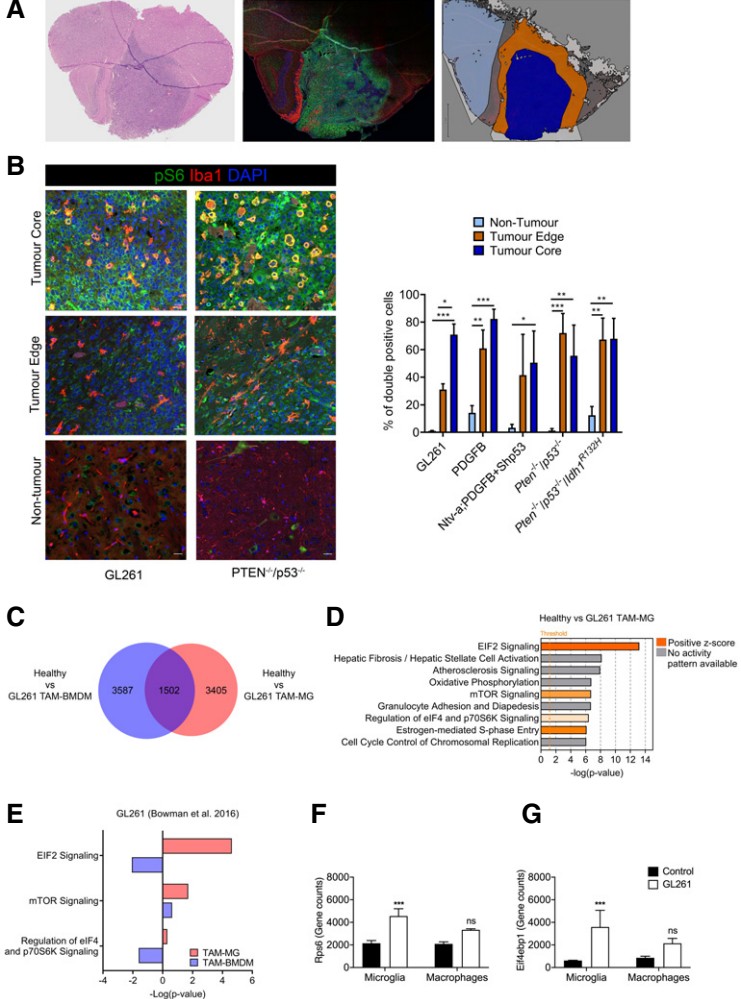

**Figure 1. Upregulation of mTOR signalling in TAM-MG but not TAM-BMDM in GBM mouse models.**

A  Representative images of a PDGFB tumour: H&E on the right to identify the tumour, an adjacent section stained for Iba1 (red), p-S6 (green) and DAPI (blue) at the centre and the region selection for quantification on the left (non-tumour in light blue, tumour edge in orange and tumour core in dark blue).

B  Representative images of core, edge and adjacent non-tumour brain tissue for Iba1, p-S6 and DAPI staining in $Pten^{-/-}p53^{-/-}$ (right) and GL261 (left) tumours. Percentage of Iba1$^+$ cells co-expressing p-S6 in the three defined regions. Five high-grade glioma models were analysed—GL261 ($n = 3$), PDGFB ($n = 3$), Ntv-a; PDGFB+shp53 ($n = 2$), $Pten^{-/-}p53^{-/-}$ ($n = 3$) and $Pten^{-/-}p53^{-/-}Idh1^{mut}$ ($n = 3$) (mean ± SEM; two-way ANOVA Tukey test).

C  Venn diagram identifying significantly deregulated genes in healthy versus GL261 TAM-MG and/or TAM-BMDM (GSE68376 dataset).

D  Top-most deregulated canonical pathways in GL261 TAM-MG, as identified by the IPA software. Threshold indicates $P \leq 0.05$. Z-score indicates the orientation of the deregulation. Ratio indicates the number of deregulated genes in the pathway.

E  mTOR-related deregulated canonical pathways in GL261 TAM-MG and TAM-BMDM, as identified by the IPA software.

F, G  Expression levels of (F) Rps6 and (G) Eif4ebp1 in healthy versus GL261 microglia and macrophage ($n = 3$; mean ± SEM, likelihood ratio test in edgeR).

Data information: *$P \leq 0.05$, **$P \leq 0.01$, ***$P \leq 0.001$.

---

significantly deregulated pathways identified by ingenuity pathway analysis (IPA; Kramer *et al*, 2014) in TAM-MG included mTOR signalling and mTOR-related signalling pathways: EIF2 signalling, regulation of eIF4 and p70S6K signalling, and mTOR signalling (Fig 1D). These pathways, as indicated by positive Z-score, were predicted to be activated (Fig 1D, Appendix Fig S1A and Dataset EV1). Importantly, these pathways were not detected as enriched when looking at the 5,189 deregulated genes in TAM-BMDM (Fig 1E, Appendix Fig S1B), and upregulation of RPS6 and EI4EBP1 was detected in TAM-MG but not in TAM-BMDM compared to controls (Fig 1F and G).

These results show an increase in mTOR activity in TAM in several mouse models of GBM, which is specific to TAM-MG and not observed in TAM-BMDM in the GL261 model.

### Glioblastoma initiating cells increase mTOR signalling via PI3K/AKT axis in tumour-conditioned microglia but not BMDM

Next, we asked how mTOR deregulation occurred in TAM-MG and whether glioblastoma initiating cells (GIC) could play a role, considering that they secrete growth and inflammatory factors that could potentially stimulate mTOR signalling in microglia (Okawa *et al*,

2017). We tested this hypothesis in an *in vitro* setting, where primary microglia and bone marrow-derived macrophages (BMDM), harvested from neonatal and 3-month-old C57BL/6 mice, were conditioned with the supernatant from different primary patient-derived GIC lines. Conditioned media was obtained from GL261 (GL261-CM) and primary $Pten^{-/-};p53^{-/-}$ mGIC cultures (mGIC$^{Pten-/-;p53-/-}$-CM), two models with increased mTOR signalling in TAM-MG *in vivo* (Fig 1B–G). The secretome of mouse neural stem cells (mNSC-CM) derived from syngeneic mice was used as a control (Fig 2A). Unconditioned microglia and BMDM cultures were also used as controls (Fig 2A).

mTORC1 specifically phosphorylates S6 at S240/244 and 4EBP1 at T37/46. Microglia conditioned with mGIC$^{Pten-/-;p53-/-}$-CM or GL261-CM showed a significant increase in phosphorylation at both of these sites (Figs 2B and D, and EV1A), when compared to cultures treated with mNSC-CM or unconditioned media, supporting the notion that the phenotype is consistent across glioblastoma models. Conversely, BMDM displayed no difference in p-S6 or p-4EBP1 levels in cultures treated with mGIC$^{Pten-/-;p53-/-}$-CM or GL261-CM, compared to mNSC-CM or unconditioned media (Fig EV1C and E). Flow cytometry analysis of these cultures confirmed the increase in p-S6 (Figs 2C and EV1B) and p-4EBP1 (Figs 2E and EV1B) levels in mGIC$^{Pten-/-;p53-/-}$-CM-treated microglia but not BMDM (Fig EV1F) compared to the mNSC-CM treatment (normalised to unconditioned cultures). Increased phosphorylation of the upstream regulator AKT at S473 sites in microglia (Figs 2F and EV1A) but not BMDM (Fig EV1C and E) treated with mGIC$^{Pten-/-;p53-/-}$-CM or GL261-CM demonstrated increased activity of mTORC2 signalling. Treatment with Torin (inhibitor of mTORC1 and mTORC2) resulted in significant reduction p-AKT (S473) in tumour-conditioned microglia (Figs 2F and EV1A), a finding which was validated by FACS (Figs 2G and EV1B and F). Increased phosphorylation of AKT at T308 site was also detected (Fig 2H). Treatment with the PI3K inhibitor (LY294002) resulted in significant reduction in p-AKT (T308) and p-S6 (S240/244) in mGIC$^{Pten-/-;p53-/-}$-CM and GL261-conditioned microglia (Figs 2H and I, and EV1A and B) but not in BMDM (Fig EV1D–F), thereby confirming activation of mTOR signalling via PI3K/AKT in tumour-conditioned microglia.

These results show that factors secreted by mGIC$^{Pten-/-;p53-/-}$ and GL261 cultures upregulate mTORC1 and mTORC2 signalling via the PI3K/AKT axes in microglia but not in BMDM.

## Genetic inhibition of mTORC1 signalling in Cx3cr1$^+$ TAM reduces tumour growth and increases survival

To investigate the functional role of activated mTOR signalling in TAM-MG, genetic inhibition was established in these cells in mice recipient of GL261 GBM allografts (Fig 3A). The GL261 cell line was chosen to generate fast-growing orthotopic syngeneic GBM models in immunocompetent mice. A mouse line with a floxed exon 3 of the *Rheb1* gene (*Rheb1*$^{fl/fl}$), a key effector of mTOR, was chosen to inactivate the pathway in TAM. Genetic modulation of mTOR signalling *in vivo* was achieved by crossing the *Rheb1*$^{fl/fl}$ mice with *Cx3cr1*-Cre$^{ERT2}$ knock-in mice, resulting in deletion of *Rheb1* in microglia upon tamoxifen-induced Cre expression. Three weeks after tamoxifen injection, GL261 tumour cells were injected intracerebrally in mutant animals as well as in

controls lacking the Cre construct but which also had received tamoxifen treatment (Fig 3A). Mice were culled when symptomatic and a longer survival was observed for the *Cx3cr1-Rheb1*$^{\Delta/\Delta}$ mice as compared to the *Rheb1*$^{fl/fl}$ mice (Fig 3B). An independent cohort of allografted mice was generated and imaged 20 days post-tumour initiation by MRI. In this cohort, measurement of tumour volume confirmed that tumours were significantly smaller in the *Cx3cr1-Rheb1*$^{\Delta/\Delta}$ compared to *Rheb1*$^{fl/fl}$ mice (Fig 3C). The experiment was terminated 25 days post-tumour initiation and the brains either processed for histology (*n* = 5) or analysed by flow cytometry (*n* = 6). Histological features were those of a highly cellular glial tumour with prominent nuclear pleomorphism, brisk mitotic activity and multifocal microvascular proliferations (Fig 3D). While no histological differences were noted between the two genotypes (Fig 3D), the numbers of GL261 GFP$^+$ tumour cells were lower in the *Cx3cr1-Rheb1*$^{\Delta/\Delta}$ tumours (Fig 3E), consistent with the reduced tumour volume observed by MRI.

Overall 98% of TAM-MG were positive for YFP, the expression of which was dependent on Cre expression in *Cx3cr1-Rheb1*$^{\Delta/\Delta}$ mice (Fig EV2A–C). Additionally, 35% of TAM-BMDM expressed YFP (Fig EV2A–C), in accordance with the known expression pattern of the *Cx3cr1* promoter in these tumours (Bowman *et al*, 2016). mTOR inhibition was confirmed in the tumours by assessing the expression levels of p-S6 in P2RY12$^+$ CD49d$^-$ TAM-MG and P2RY12$^-$ CD49d$^+$ TAM-BMDM by flow cytometry. We observed that p-S6 baseline levels in *Rheb1*$^{fl/fl}$ tumours were higher in TAM-MG than TAM-BMDM, and these were significantly reduced in TAM-MG but not in TAM-BMDM in *Cx3cr1-Rheb1*$^{\Delta/\Delta}$ tumours (Fig EV2A–C). A clear reduction in the number of Iba1$^+$ p-S6$^+$ cells was also seen in the *Cx3cr1-Rheb1*$^{\Delta/\Delta}$ versus *Rheb1*$^{fl/fl}$ tumour tissues (Fig EV2D and E).

The above data show that inactivation of mTORC1 in TAM reduces tumour growth and increases survival in a GBM mouse model.

## Genetic inhibition of mTORC1 signalling in TAM affects the innate/adaptive immune system crosstalk in GL261 tumours

To further characterise the phenotype observed in *Cx3cr1-Rheb1*$^{\Delta/\Delta}$ GL261 model, we analysed the transcriptome of these tumours. Principal component analysis revealed distinct clustering of *Cx3cr1-Rheb1*$^{\Delta/\Delta}$ GL261 allografts (*n* = 3) from *Rheb1*$^{fl/fl}$ tumours (*n* = 3) (Fig EV3A). 425 genes were identified as differentially expressed between the two genotypes, the majority of which (302) were downregulated in *Cx3cr1-Rheb1*$^{\Delta/\Delta}$ tumours, with only 123 upregulated (Fig EV3B). IPA analysis identified pathways associated with antigen presentation and innate to adaptive immune cell communication as enriched, including dendritic cell maturation, antigen presentation pathway, communication between innate and adaptive immune cells, iCOS-iCOSL signalling, OX40 signalling, CD28 signalling and crosstalk between dendritic cells and natural killer cells (Fig 4A). Csf1r and Csf1 were amongst the downregulated genes, as well as markers of TAM including Aif1 (Iba1) and Itgam (Cd11b), suggesting a change in the activity profile of TAM as a result of mTOR inhibition or a change in the immune composition of infiltrating immune cells (Fig EV3C). Moreover, pathways associated with cytokine signalling were also detected,

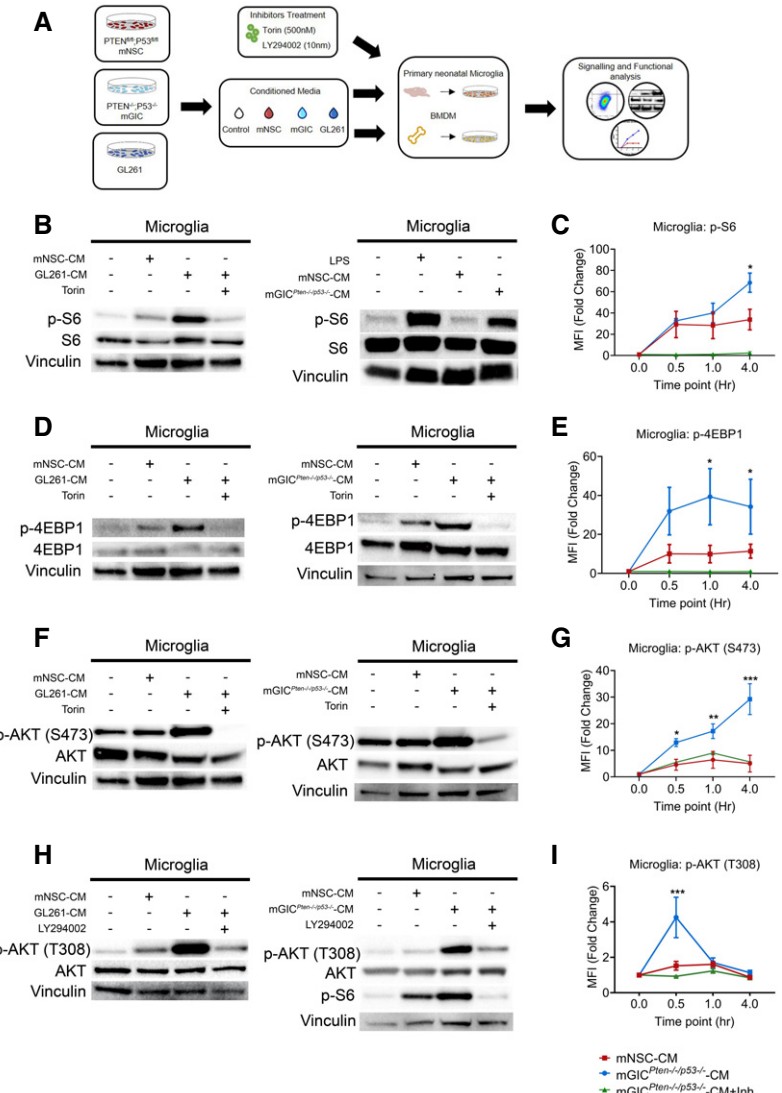

**Figure 2. Microglia and BMDM are differently conditioned by mGIC.**

A    Schematic of the *in vitro* model whereby microglia and BMDM were pretreated with Torin, LY294002 as indicated and stimulated with mGL261, mGIC$^{Pten-/-;p53-/-}$-CM or mNSC-CM.

B–I   Signalling was analysed in microglia by immunoblotting of whole cell lysates collected at 4 h (B, D and F) and 0.5 h (H) and normalised against non-phosphorylated protein and vinculin analysed on the same blot. Flow cytometry analysis was carried out in microglia for (C) p-S6 S240/244; (E) p-4EBP1 T37/46; (G) p-AKT S473; and (I) p-AKT T308. Each treatment (mNSC-CM, $Pten-/-p53-/-$ mGIC-CM, $Pten-/-p53-/-$ mGIC-CM+Torin, mGIC-CM+LY) was normalised to unconditioned control (n = 3; mean ± SEM, two-way ANOVA Tukey test). *$P \leq 0.05$, **$P \leq 0.01$, ***$P \leq 0.001$ comparing mGIC$^{Pten-/-;p53-/-}$-CM versus mGIC$^{Pten-/-;p53-/-}$-CM +inhibitor.

Source data are available online for this figure.

including IL-4, IL-10, IL-6, IL-2, IL12 signalling, STAT3 pathway, NF-κB signalling, iNOS signalling, interferon signalling and Toll-like receptor (TLR) signalling (Fig 4A). Amongst the top scoring pathways, several were associated with regulation of T-cell signalling, differentiation and activation—including T helper cell differentiation, Th1 and Th2 activation pathway, T-cell exhaustion signalling, PD-1 and PD-L1 cancer immunotherapy pathway (Fig 4A, Dataset EV2). Amongst the differentially expressed genes, CD274 (PD-L1) was downregulated in the $Cx3cr1$-$Rheb1^{\Delta/\Delta}$ compared to $Rheb1^{fl/fl}$ GL261 tumours, further highlighting a

reduced level of T-cell inhibition via checkpoint inhibitors in the TME (Fig EV3C). These data show that mTOR inhibition in Cx3cr1$^+$ TAM reshapes the immune landscape of the TME by influencing the expression of inflammatory mediators as well as the crosstalk between the innate and adaptive immune system.

The deregulation of these immune-related pathways prompted us to further study the composition of the immune infiltrates in the TME of $Cx3cr1$-$Rheb1^{\Delta/\Delta}$ GL261 allografted tumours. We performed cell-type identification by estimating relative subsets of RNA transcripts (CIBERSORT) (Newman *et al*, 2015), a

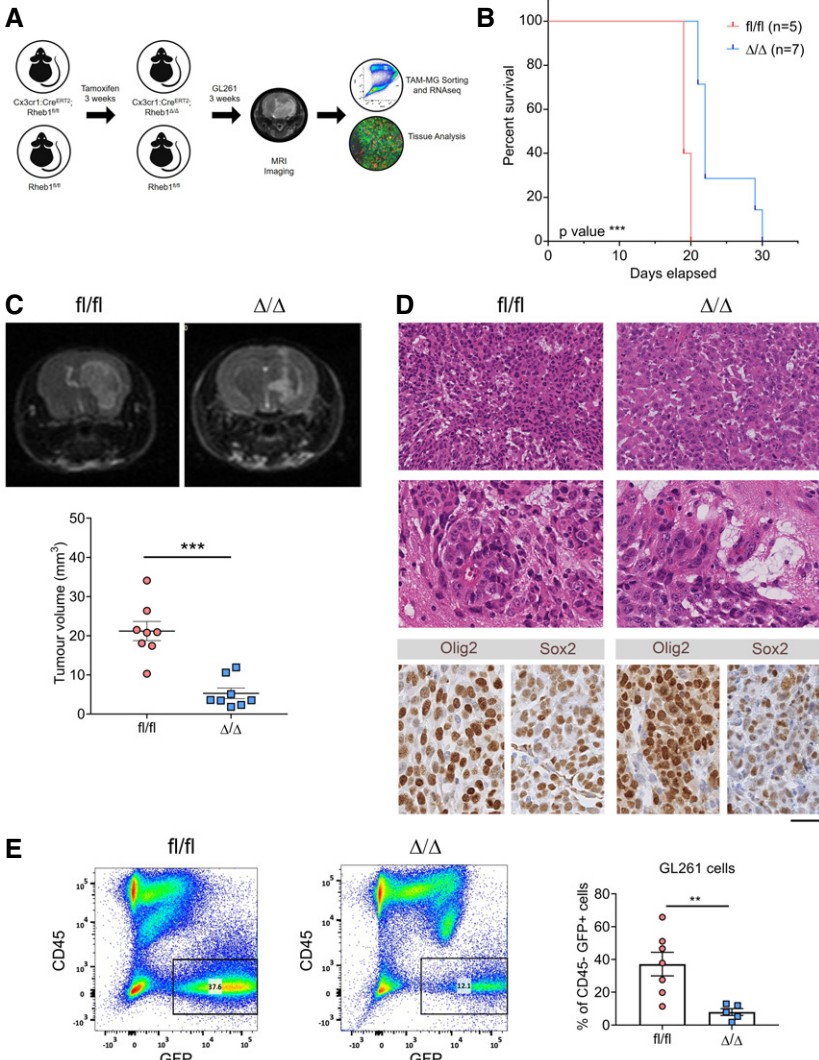

**Figure 3. Genetic inhibition of mTORC1 signalling in Cx3cr1$^+$ TAM impacts tumour growth and survival.**

A  Schematic of the generation and analysis of the *Cx3cr1*-Cre;*Rheb1*-loxp GL261 model.

B  Survival analysis for *Cx3cr1-Rheb1*$^{\Delta/\Delta}$ (*n* = 7) and *Rheb1*$^{fl/fl}$ (*n* = 7) mice. Chi-square test.

C  Representative images and quantification of the tumour volume with MRI of *Rheb1*$^{fl/fl}$ (*n* = 8) compared to *Cx3cr1-Rheb1*$^{\Delta/\Delta}$ (*n* = 8) mice (mean ± SEM; unpaired parametric *t*-test).

D  H&E staining showing representative histological features (overview and high magnification of microvascular proliferation) of *Rheb1*$^{fl/fl}$ and *Cx3cr1-Rheb1*$^{\Delta/\Delta}$ GL261 tumours. Scale bar is 125 µm (top, H&E) and 80 µm (all other images)

E  Percentage of GFP$^+$ CD45$^-$ GL261 tumour cells in *Cx3cr1-Rheb1*$^{\Delta/\Delta}$ (*n* = 5) and *Rheb1*$^{fl/fl}$ (*n* = 7) tumours, with representative FACS plot (mean ± SEM; unpaired parametric *t*-test).

Data information: **$P \le 0.01$, and ***$P \le 0.001$.

---

computational approach which accurately predicts the relative fraction of different cell subsets from gene expression profiles of complex tissues (cibersort.stanford.edu). When comparing the cellular fractions between the *Cx3cr1-Rheb1*$^{\Delta/\Delta}$ and *Rheb1*$^{fl/fl}$ GL261 tumours, a significant increase in CD8 cytotoxic T lymphocytes (CTL) and CD4 helper T (Th) cells was detected, with no changes in regulatory T (Treg) cells (Fig EV3D). The monocyte cell fraction was also significantly increased. However, the analysis did not allow us to differentiate between monocytes, TAM-BMDM and TAM-MG. To validate the CIBERSORT findings

and further characterise the TAM population, the immune composition of *Cx3cr1-Rheb1*$^{\Delta/\Delta}$ *Rheb1*$^{fl/fl}$ GL261 tumours was analysed by flow cytometry. This revealed a shift in the TAM population in the *Cx3cr1-Rheb1*$^{\Delta/\Delta}$ tumours, with a significant decrease in TAM-MG (CD45$^+$ P2RY12$^+$ CD49d$^-$), while significantly higher numbers of TAM-BMDM (CD45$^+$ P2RY12$^-$ CD49d$^+$) were observed (Fig 4B). The lymphocyte fraction showed an increase in CD8 CTL (CD45$^+$ CD3$^+$ CD8$^+$) and CD4 Th cells (CD45$^+$ CD3$^+$ CD4$^+$) in the *Cx3cr1-Rheb1*$^{\Delta/\Delta}$ tumours (Fig 4C). Quantification of immunolabelled cells *in situ* confirmed

increased CD8$^+$ CTLs and CD4$^+$ Th cells, with FoxP3$^+$ Treg cell numbers remaining unchanged in the *Cx3cr1-Rheb1*$^{\Delta/\Delta}$ tumours (Fig 4D).

Taken together, these data raise the possibility that downregulation of mTOR signalling in TAM-MG impairs proliferation or recruitment of these cells while increasing that of peripheral immune

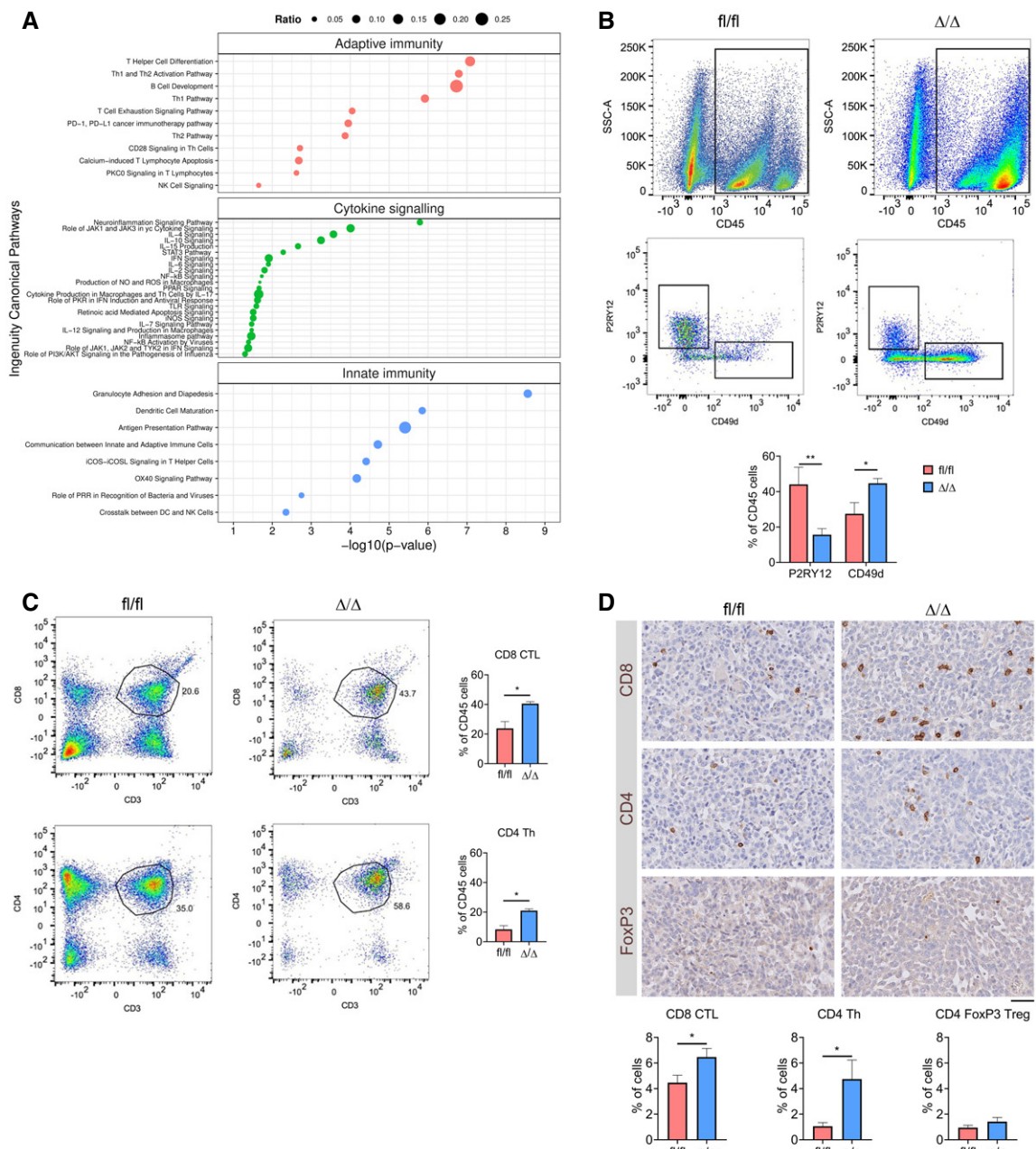

**Figure 4. Genetic inhibition of mTORC1 signalling in TAM affects the immune profile of GL261 tumours.**

A Significantly deregulated canonical pathways in *Cx3cr1-Rheb1*$^{\Delta/\Delta}$ versus *Rheb1*$^{fl/fl}$ GL261 tumours, as identified by the IPA software from differentially regulated genes and divided into three categories: adaptive immunity, cytokine signalling and innate immunity. Size of bubbles is indicative of ratio of differentially regulated genes. The x-axis represents $-\log_{10}$ (P-value) with a threshold of $P \leq 0.05$ applied.

B Levels of TAM-MG (P2RY12$^+$ CD49d$^-$) and TAM-BMDM (P2RY12$^-$ CD49d$^+$) gated from CD45$^+$ population in GL261 tumours from *Rheb1*$^{fl/fl}$ (n = 4) and *Cx3cr1-Rheb1*$^{\Delta/\Delta}$ (n = 5) mice by flow cytometry (representative flow cytometry plots on top and quantification at the bottom; mean ± SEM; two-way ANOVA Tukey test).

C Percentage of CD8 CTL (top) and CD4 Th (CD4) cells in GL261 tumours from *Rheb1*$^{fl/fl}$ (n = 4) and *Cx3cr1-Rheb1*$^{\Delta/\Delta}$ (n = 5) mice assessed by flow cytometry (representative flow cytometry plots, left and quantification, right; mean ± SEM; unpaired t-test).

D Tumour tissues from *Rheb1*$^{fl/fl}$ (n = 4) and *Cx3cr1-Rheb1*$^{\Delta/\Delta}$ (n = 4) were stained for CD8, CD4 and FoxP3. DAB staining was quantified as % of positive cells using Definiens software. Representative images of the stained tissues are shown (mean ± SEM; unpaired parametric t-test). Scale bar is 100 µm.

Data information: *$P \leq 0.05$ and **$P \leq 0.01$.

infiltrates, including both monocyte-derived macrophages and effector T cells.

## Genetic inhibition of mTORC1 signalling in TAM increases proliferation and effector function of CD4 and CD8 T cells in GL261 tumours

Having identified changes in the tumour immune landscape, we next characterised the activity profile of the infiltrating lymphocytes. To this end, we comparatively analysed immune cell fractions, as determined by CIBERSORT, with the enrichment score of specific immune pathways identified in bulk tumour tissue (Fig 5A, Table EV1). A differential correlation score was calculated between the $Cx3cr1$-$Rheb1^{\Delta/\Delta}$ and $Rheb1^{fl/fl}$ GL261 tumours (Fig 5A). As expected, mTOR signalling negatively correlated with TAM with the $Cx3cr1$-$Rheb1^{\Delta/\Delta}$ background (Appendix Fig S2). Moreover, the negative regulation of lymphocytes pathway, which also negatively correlated with neutrophils and TAM, clustered with mTOR signalling and not with other pathways. This suggests a shift in the regulation of T-cell activity by TAM with an increased stimulatory capacity driven by mTOR inhibition. Additional support for this interpretation is provided by the observation that the negative regulation of lymphocytes pathway negatively correlated with the CD8 CTL, Tregs and CD4 lymphocyte fractions (Fig 5B, Pearson correlation scores can be found in Table EV2), while the antigen presentation pathway correlated positively. These findings suggest a change in the regulation of T-cell activity in the TME upon downregulation of mTOR in TAM from an immunosuppressive to increased effector function.

To validate the CD4/CD8 T-cell enhanced effector function, as suggested by the *in silico* analysis, we analysed the expression of IFNγ, perforin and granzyme b in the tumour-infiltrating lymphocyte populations by flow cytometry. An increased expression of perforin and IFNγ was detected in CD4 Th cells (Fig 5C), and an increase of perforin and granzyme b was detected in CD8 CTL (Fig 5D). Furthermore, to assess whether changes in T-cell levels in TME of $Cx3cr1$-$Rheb1^{\Delta/\Delta}$ tumours were due to infiltration and/or proliferation, we examined the expression of Ki67 and of adhesion molecules CD44 and CD62L. CD8 CTL and CD4 Th cells displayed increased proliferation with significantly higher Ki67 levels (Fig 5C and D). Moreover, both T-cell populations displayed an increase in CD44 expression (Fig 5E) that correlated with a decreased CD62L expression (Fig 5F) in $Cx3cr1$-$Rheb1^{\Delta/\Delta}$ compared to $Rheb1^{fl/fl}$ tumours. This indicates an increase in infiltration as well as antigenic stimulation of T cells, with an increase in memory/effector versus naïve T cells, which results from mTOR inhibition in $Cx3cr1^+$ TAM.

These data are in keeping with TAM inhibiting infiltration, proliferation and function of effector T cells via the mTOR pathway, in a GBM mouse model.

## mTORC1 inhibition in TAM-MG induces a pro-inflammatory tumour microenvironment in GL261 allografts

To understand the molecular mechanism underpinning the change in T-cell infiltration, proliferation and effector activity observed in $Cx3cr1$-$Rheb1^{\Delta/\Delta}$ GL261 allografted tumours, we analysed the transcriptomic profile of $Rheb1^{\Delta/\Delta}$ TAM-MG ($n = 3$) as compared to

$Rheb1^{fl/fl}$ TAM-MG ($n = 2$). TAM-MG were isolated from the tumour bulk as the $CD45^{LOW}$ $CD11b^+$ population by FACS (Fig EV2A). As expected, a high enrichment score for the TAM-MG signature (Bowman *et al*, 2016) was detected in these cells, confirming the purity of the sorted TAM-MG population (Fig EV4A). Principal component analysis revealed distinct clustering of $Rheb1^{\Delta/\Delta}$ from $Rheb1^{fl/fl}$ TAM-MG (Fig EV3A) with 988 genes differentially expressed between the two genotypes: 823 genes were upregulated in $Rheb1^{\Delta/\Delta}$ TAM-MG, and only 165 were downregulated (Fig EV4B). IPA analysis identified enriched pathways associated with inflammatory signalling, including cytokine signalling pathways, such as IL-12, IL-6 and IL-8 signalling; pathways linked to production of nitric oxide (NO) and reactive oxygen species (ROS); receptor signalling pathways, such as pattern recognition receptors (PRR) and Toll-like receptors (TLR); and pathways linked to signalling in granulocytes and antigen-presenting cells (APCs), including granulocyte adhesion and diapedesis, dendritic cell maturation and granzyme A signalling (Fig 6A, Dataset EV2). Amongst the top scoring pathways, several were associated with the regulation of T-cell signalling and differentiation, in agreement with our observations from the bulk tissue transcriptome. When examining the differentially deregulated genes identified in these pathways, the majority of genes across all pathways were found to be upregulated (Fig 6A), raising the possibility that these pathways could be under co-ordinated regulation.

To characterise the mechanistic basis for the increased number of T effector cells, we looked more closely at the differentially regulated genes in $Rheb1^{\Delta/\Delta}$ TAM-MG. Amongst downregulated genes, the chemokines Ccl5 and Cxcl13 were most frequently associated with the enriched canonical pathways. Amongst upregulated genes, NF-κB signalling (Nf-κb1), PI3K class I and II signalling (Pik3cg, Pik3c2b) and type I interferon signalling (Ifnar1, Tlr4) were identified as most frequently deregulated (Fig 6B). When looking at upstream regulators (predicted by IPA as potentially responsible for the observed transcriptomic profile), the majority were associated with interferon signalling (Fig 6C). These included interferon-regulatory factors (Irf), members of the Stat family of transcription factors and interferon receptors. Furthermore, Isg15 and Usp18, inhibitors of type I interferon signalling, were amongst the few significantly downregulated genes (Fig EV4C). Importantly, a predicted upregulation of NF-κB signalling was detectable (Fig EV4D). Interestingly, several of the signalling pathways identified as deregulated in $Rheb1^{\Delta/\Delta}$ TAM-MG were found to be significantly deregulated in GL261 TAM-MG but not in TAM-BMDM, including IFN signalling, NF-κB and STAT3 signalling (Fig EV4E).

To gain further support for a regulatory role of mTOR signalling on these signalling pathways, a Pearson correlation analysis was run between the ssGSEA enrichment score of deregulated pathways in the $Rheb1^{\Delta/\Delta}$ TAM-MG. The enrichment of mTOR signalling positively correlated with the enrichment of the negative regulation of lymphocytes signature, while they both negatively correlated with NF-κB and IFNγ signalling as well as leucocyte differentiation, Th1 and Th2 differentiation and T-cell chemotaxis signatures (Fig 6D).

Therefore, our data suggest that inhibition of mTOR in TAM-MG remodels the immune composition of the microenvironment and increases the presence of effector T cells through the

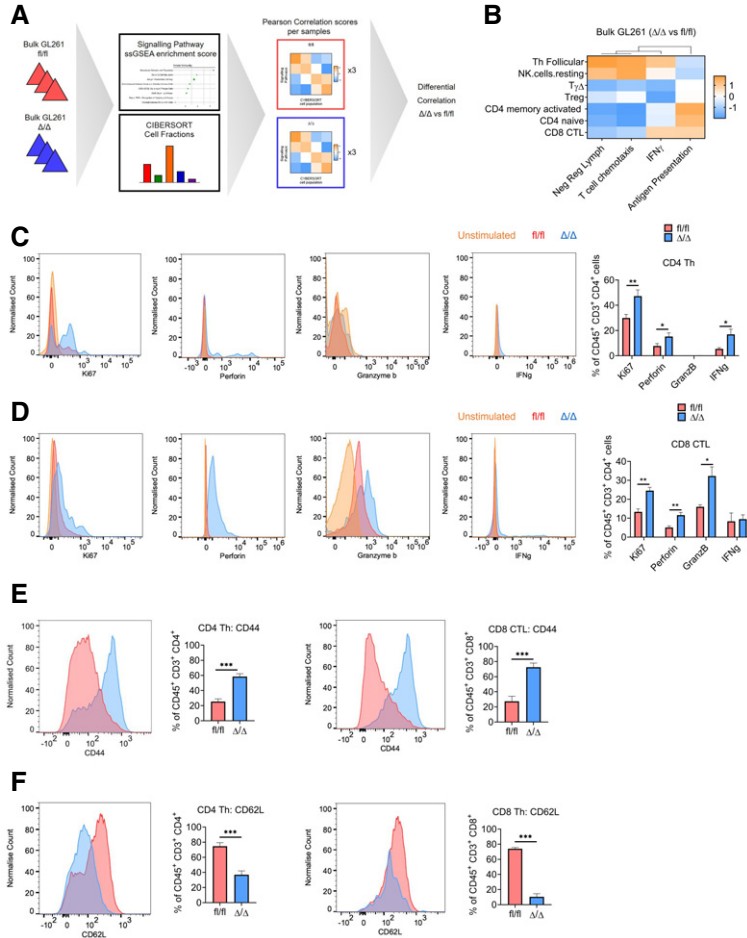

**Figure 5. Genetic inhibition of mTORC1 signalling in TAM affects the immune reactivity of T cells.**

A Schematic of the analysis of bulk GL261 RNA-Seq data from *Cx3cr1-Rheb1*$^{\Delta/\Delta}$ (*n* = 3) and *Rheb1*$^{fl/fl}$ (*n* = 3) mice. ssGSEA enrichment scores for signalling pathway signatures and CIBERSORT cell fractions were calculated for each sample. Pearson correlation score was then calculated comparing signalling pathway enrichment scores and CIBERSORT cell fractions. Finally, a differential correlation analysis was run between the *Cx3cr1-Rheb1*$^{\Delta/\Delta}$ (*n* = 3) and *Rheb1*$^{fl/fl}$ (*n* = 3) samples.

B Heatmap of the differential correlation analysis between CIBERSORT cell fractions and signatures enrichment scores (ssGSEA) for lymphocytes in the *Cx3cr1-Rheb1*$^{\Delta/\Delta}$ (*n* = 3) vs. *Rheb1*$^{fl/fl}$ (*n* = 3) GL261 tumours.

C, D CD4 Th cells (C) and CD8 CTL cells (D) expression of Ki67, perforin, granzyme b and IFNγ, in *Cx3cr1-Rheb1*$^{fl/fl}$ (*n* = 5) and *Rheb1*$^{\Delta/\Delta}$ (*n* = 5) GL261 tumours. Representative FACS plots show control samples (tumour-derived cells stimulated in culture in the absence of protein inhibitor cocktail—in orange) compared to stimulated cells derived from *Rheb1*$^{fl/fl}$ (fl/fl, red) and *Cx3cr1-Rheb1*$^{\Delta/\Delta}$ (Δ/Δ, blue) GL261 tumours (cells stimulated in culture with protein inhibitor cocktail) (mean ± SEM; two-way ANOVA Tukey test).

E, F Expression of CD44 (E) and CD62L (F) by CD4 Th and CD8 CTL in *Cx3cr1-Rheb1*$^{fl/fl}$ (*n* = 5; fl/fl, red) and *Rheb1*$^{\Delta/\Delta}$ (*n* = 5; Δ/Δ, blue) GL261 tumours with representative FACS plots show control samples (mean ± SEM; two-way ANOVA Tukey test).

Data information: \*$P \leq 0.05$, \*\*$P \leq 0.01$, \*\*\*$P \leq 0.001$.

---

upregulation of pro-inflammatory pathways, including IFNγ and NF-κB signalling.

### mTOR-dependent expression of inflammatory cytokines in tumour-conditioned microglia promotes an anti-inflammatory phenotype via the regulation of STAT3 and NF-κB transcription factors

Next, we set out to validate the *in silico* prediction from the *in vivo* transcriptomic profile of *Rheb1*$^{\Delta/\Delta}$ TAM-MG that mTOR signalling negatively regulates NF-κB in TAM-MG and is responsible for the immunosuppressed phenotype in GBM. In addition, to test for a possible contribution of the STAT family of transcription factors, we assessed STAT3 activity. STAT3 has been previously reported as deregulated in GBM TAM and associated with expression of anti-inflammatory cytokines, contributing to an immunosuppressed TME (Wu *et al*, 2010; West *et al*, 2018). Moreover, it has been shown to regulate the inflammatory activity of monocytes in response to infection in an mTOR-dependent manner (Weichhart *et al*, 2008).

We used our *in vitro* experimental system (Fig 2A) to assess whether the mTOR-dependent activity of these transcription factors was responsible for the pro-inflammatory profile of TAM-MG. While no changes in p-NF-κB (p-P65) levels were detected in tumour-conditioned BMDM (using mGIC$^{Pten-/-;p53-/-}$-CM and GL261-CM)

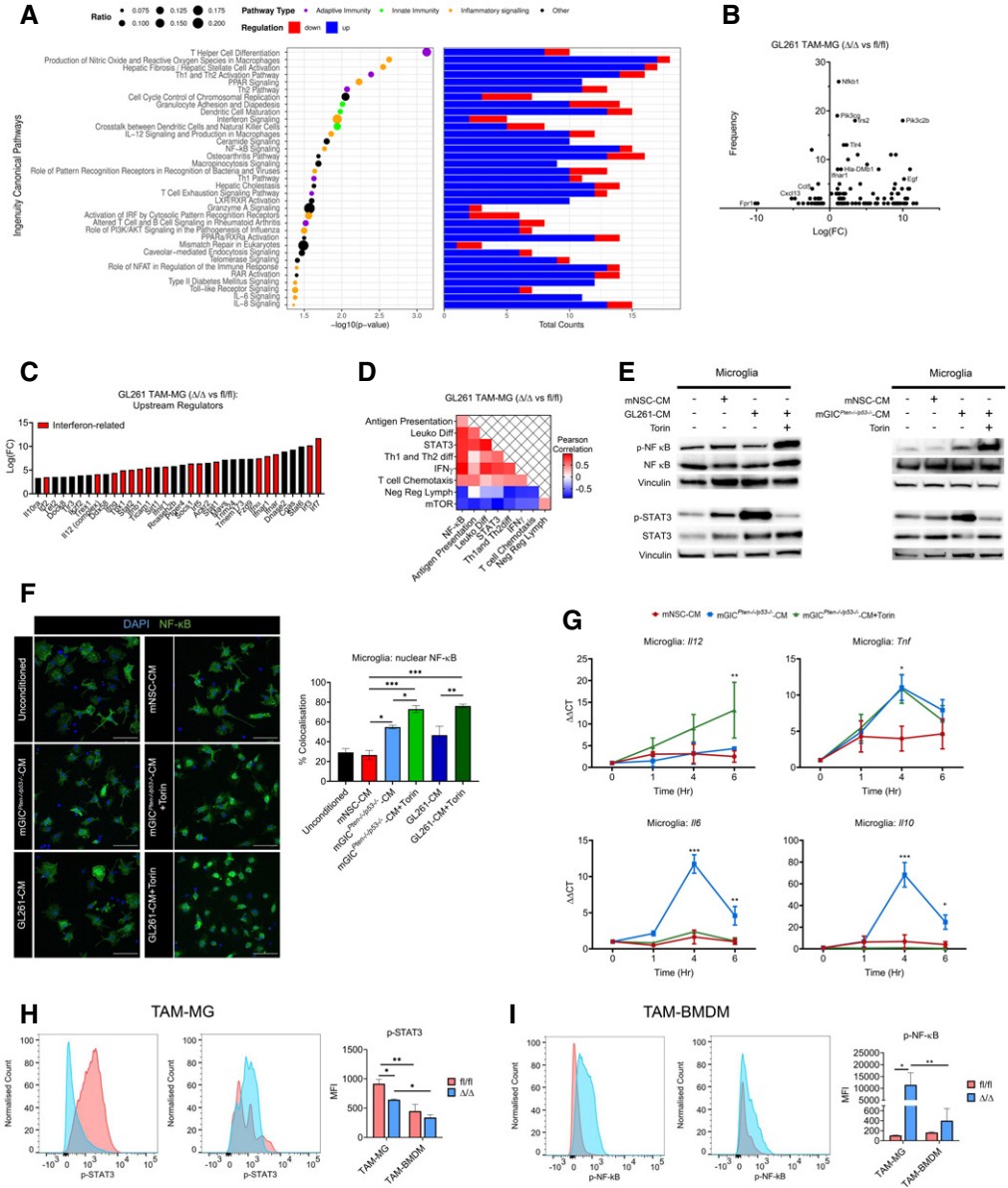

**Figure 6. mTOR inhibition in TAM-MG induces a pro-inflammatory tumour microenvironment via the regulation of STAT3 and NF-κB transcription factors.**

A Significantly deregulated canonical pathways in GL261 *Rheb1*^fl/fl versus *Rheb1*^Δ/Δ TAM-MG, as identified by the IPA software from differentially regulated genes. Size of bubbles is indicative of ratio of differentially regulated genes. The *x*-axis represents −log$_{10}$ (*P*-value) with a threshold of $P \leq 0.05$ applied. On the right, number of up and down regulated genes in each identified pathway is indicated in blue and red respectively.

B Frequency of occurrence of differentially regulated genes across significantly deregulated canonical pathways compared to the log fold change (Log(FC)).

C Upstream regulators as identified by the IPA software from differentially regulated genes.

D Heatmap of Pearson correlation analysis between ssGSEA signatures enrichment scores in the *Rheb1*^fl/fl versus *Cx3cr1-Rheb1*^Δ/Δ GL261 tumours.

E Microglia were pretreated with Torin or medium as indicated and then stimulated with mNSC-CM, mGIC^Pten−/−;p53−/−-CM or GL261-CM. p-NF-κB (p-P65) and p-STAT3 were analysed by immunoblotting of whole cell lysates collected at 4 h.

F Immunofluorescence images of NF-κB (P65) in conditioned microglia. Quantification of nuclear translocation of NF-κB from staining. Units represent the % of voxels in the NF-κB channel colocalised with DAPI (*n* = 3; mean ± SEM; one-way ANOVA Tukey test—right). Scale bar is 50 µm.

G Production of Il-12 (Il-12p40), Tnf, Il6 and Il10 by conditioned microglia was determined by qPCR. Each treatment was normalised to housekeeping gene and the unconditioned control (*n* = 3; mean ± SEM; two-way ANOVA Tukey test).

H, I MFI levels of pSTAT3 in TAM-MG (CD45+ P2RY12+ CD49d−) (H) and p-NF-κB (p-P65) in TAM-BMDM (CD45+ P2RY12− CD49d+) (I) from *Rheb1*^fl/fl (fl/fl, blue, *n* = 4) and *Cx3cr1-Rheb1*^Δ/Δ (Δ/Δ, red, *n* = 5) GL261 tumours. Representative FACS plots are displayed for each cell type (mean ± SEM, two-way ANOVA Tukey test).

Data information: *$P \leq 0.05$, **$P \leq 0.01$, ***$P \leq 0.001$.

(Fig EV4G), we observed a slight increase phosphorylation of NF-κB (p-P65) in tumour-conditioned microglia, an effect enhanced by mTOR inhibition (Figs 6E and EV4F). To further validate the mTOR-dependent regulation of NF-κB (P65), nuclear localisation of this transcription factor was assessed in tumour-conditioned microglia and BMDM. While mGIC$^{Pten-/-;p53-/-}$-CM increased nuclear localisation of NF-κB (P65) in microglia, mGL261-CM had no effect compared to mNSC-CM (Fig 6F). However, mTOR inhibition (using Torin) combined with tumour conditioning (mGIC$^{Pten-/-;p53-/-}$-CM or GL261-CM) significantly increased the nuclear translocation of NF-κB (P65), compared to NSC-CM and tumour-conditioning alone (Fig 6F). BMDM displayed no difference in nuclear localisation of NF-κB (P65), under any condition (Fig EV4H). In keeping with this finding, IL-12 expression was increased only upon treatment with Torin. Moreover, TNF, another cytokine regulated by NF-κB, was upregulated by mGIC-CM in an mTOR-independent fashion (Fig 6G). We also observed increased phosphorylation of STAT3 in microglia treated with mGIC-CM and GL261-CM, an effect which was lost upon inhibition of mTOR signalling by Torin, as assessed by Western blotting (Figs 6E and EV4F) and flow cytometry (Fig EV4I). Once again, this effect was not observed in tumour-conditioned BMDM with mGIC$^{Pten-/-;p53-/-}$-CM or GL261-CM (Fig EV4G). Importantly, upregulation of IL-10 and IL-6 was observed in microglia treated with mGIC-CM, as predicted given the increased p-STAT3. However, it was not observed under mTOR inhibition (Fig 6G), in accordance with the expression of anti-inflammatory cytokines IL-10 and IL-6 being mediated by mTOR-dependent regulation of STAT3 activity. Moreover, P2RY12$^+$ CD49d$^-$ TAM-MG but not P2RY12$^-$ CD49d$^+$ TAM-BMDM expressed significantly lower levels of p-STAT3 (Fig 6H) and higher levels of p-NF-κB (p-P65) in *Cx3cr1-Rheb1*$^{Δ/Δ}$ allografts (Fig 6I), validating our *in vitro* findings.

In conclusion, increased phosphorylation of STAT3 in tumour-conditioned microglia upregulates the expression of IL-10 and IL-6 in an mTOR-dependent fashion with a concomitant reduction in expression of IL-12 mediated by reduced phosphorylation and nuclear translocation of NF-κB.

### Enrichment of mTOR signalling correlates with TAM-MG and a negative regulation of T cells in TCGA-GBM samples

In order to assess the translational value of our findings in human glioblastoma, we took advantage of the TCGA dataset, a publicly available database with transcriptomic data for tissue bulk from 138 IDH-wild-type GBM. To extract information specific to TAM from bulk sequencing, we carried out a correlation analysis between the mTOR pathway and TAM-MG or TAM-BMDM gene expression signatures. Using single sample gene set enrichment analysis (ssGSEA; Barbie *et al*, 2009), enrichment scores were calculated for each patient for predefined signatures of mTOR signalling, as well as TAM-MG and TAM-BMDM. TAM-MG and TAM-BMDM signatures have been previously characterised (Bowman *et al*, 2016), and the mTOR signature was obtained from the mSigDB database (Table EV1). A significant positive linear correlation was found between the enrichment score for the mTOR signature and the TAM-MG signature but not for the TAM-BMDM signature (Fig 7A, Table EV3). IDH-wild-type GBM were then further grouped according to their molecular profiles using the transcriptional classification

of Wang *et al* (2017). The positive correlation between mTOR and TAM-MG signatures was most significant in the mesenchymal subgroup and not present in the pro-neural subgroup (Fig 7A, Table EV3). These results were replicated in an additional dataset (Fig EV5A).

Next, we used the TCGA dataset to assess the impact of this mTOR signature on TAM-MG phenotype and on the immune composition of the tumours. We separated tumours into those with the signature (group 1: positive mTOR and microglia enrichment, displayed in orange) and those without the signature (group 2: displayed in green; Fig 7B). A CIBERSORT analysis was run on these two groups to identify any differences in their immune composition. The first striking result was that the two groups displayed contrary immune composition (Fig 7C), with GBM in group 1 showing more Tregs, CD4 naïve and memory resting cells and less CD8 CTL and CD4 memory-activated cells, while GBM in group 2 displayed the opposite pattern (Fig 7C). Importantly, the low level of CD8 CTL and CD4 memory-activated cells in GBM with the mTOR signature is reminiscent of our findings in GBM mouse models with the signature.

To gain additional evidence for the observed difference in immune cell composition between the tumour groups being driven by mTOR signalling in TAM-MG, we looked at the signalling activity of TAM-MG compared to TAM-BMDM in GBM from group 1. We correlated the enrichment of TAM-MG and TAM-BMDM (using the Bowman *et al* gene signature) with that of signalling pathways identified as mTOR-dependent in the mouse model, including NF-κB, STAT3, IFNγ, Th1/Th2 differentiation, T-cell chemotaxis, antigen presentation and the negative regulation of lymphocytes (Fig 7D). The mTOR pathway and the negative regulation of lymphocytes emerged as a separate cluster. In TAM-MG, the mTOR pathway and the negative regulation of lymphocytes were positively correlated, while the other pathways were negatively correlated, in accordance with our findings in mouse models (Fig 7D). While TAM-BMDM enrichment positively correlated with mTOR as well, correlation with the rest of the signatures did not follow the same pattern as observed in the mouse model, for example a negative correlation was found with the negative regulation of lymphocytes (Fig 7D).

These data confirm that a positive correlation between deregulation of mTOR signalling and TAM-MG but not TAM-BMDM is also found in human GBM. These data also show that GBM with the signature (high mTOR and microglia enrichment) display stronger depletion of activated lymphocytes compared to GBM without the signature, a phenotype potentially driven by mTOR-dependent TAM-MG activity.

### mTOR signalling is deregulated in iMGL treated with syngeneic human GIC-CM

To assess the functional relevance in human GBM of the findings in mouse models and of the *in silico* TCGA data, a GBM syngeneic-induced EPSC-derived microglia-like (iMGL) cell model was established (Fig 7E).

Two human GIC lines (hGIC) were used. They were both derived from IDH-wild-type GBM tissue according to standard protocols (Pollard *et al*, 2009) and confirmed to belong to the mesenchymal subgroup by DNA methylation array (Illumina 450 K) on bulk tumour as well as on GIC, followed by classification on the

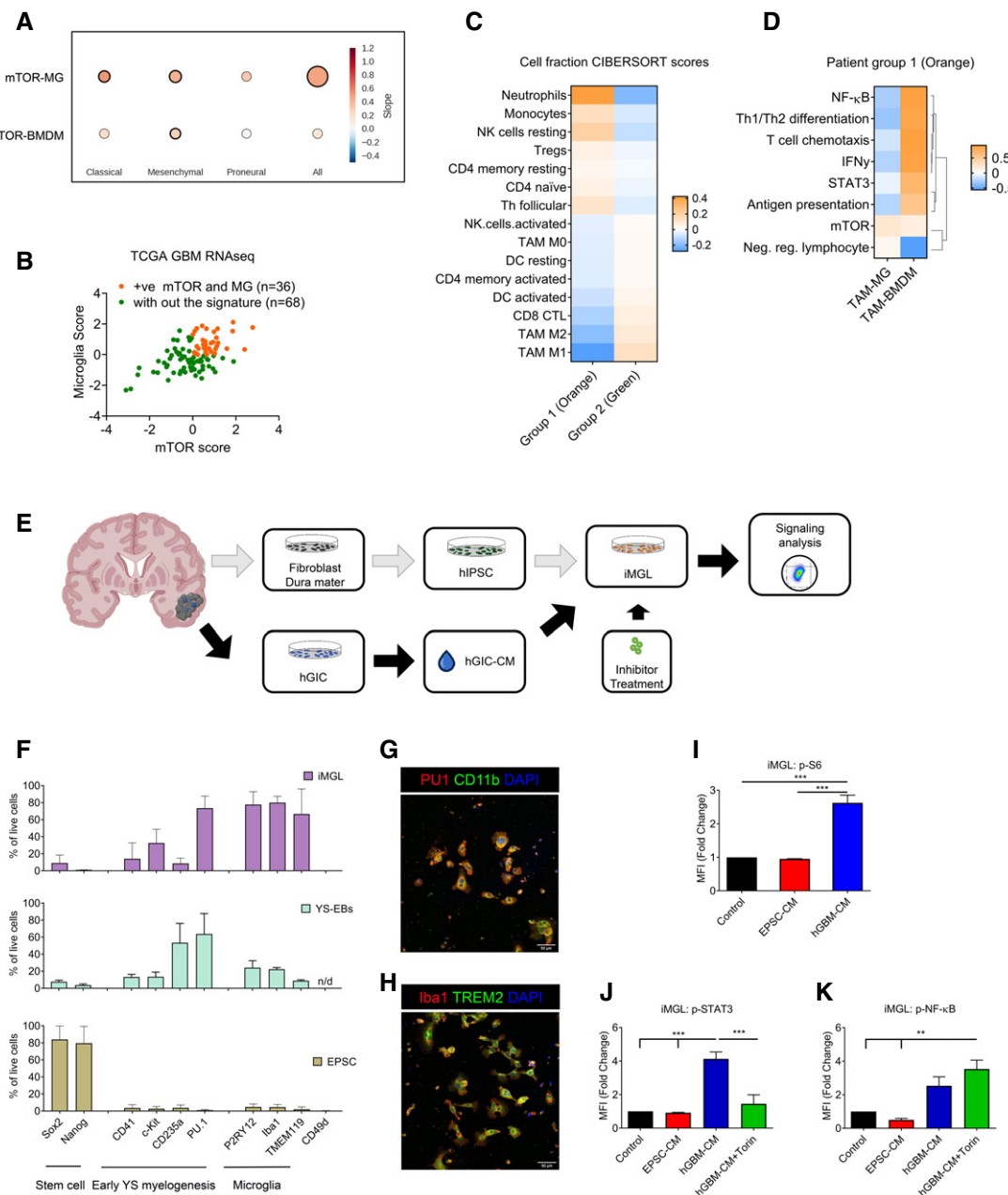

**Figure 7. mTOR signalling in TAM-MG promotes immune evasion mechanisms in human glioblastoma.**

A    Correlation between ssGSEA enrichment scores for the mTOR signature versus TAM-MG or TAM-BMDM signatures in TCGA-GBM transcriptomic data. Comparison carried out on all IDH-wild-type samples and in a subgroup-specific manner according to Wang's classifier. Size of circle is indicative of R-square value, and bold outline represents a $P \leq 0.05$.

B    Separation of IDH-wild-type GBM samples between those displaying high mTOR and microglia enrichment (+ve correlation, group 1 in orange) and those without this signature (group 2 in green).

C    CIBERSORT cell fractions calculated from the TPM of TCGA IDH-wild-type GBM samples from group 1 compared to those from group 2.

D    Heatmap of Pearson correlation analysis comparing ssGSEA enrichment scores of TAM-MG and TAM-BMDM signatures versus signalling pathway signatures, calculated from the TPM of IDH-wild-type GBM samples with the signature from group 1.

E    Schematic of the experimental setup to derive patient-matched EPSC and GIC lines.

F    Staining by flow cytometry of EPSC, YS-EBs and matched-iMGL with indicated markers ($n = 3$; mean $\pm$ SEM).

G, H  Mature iMGL were immunostained for (G) PU1 and CD11b and (H) Iba1 and TREM2. Scale bar 20 μm.

I–K  (I) P-S6, (J) p-STAT3 and (K) p-NF-κB (p-P65) were analysed by flow cytometry in iMGL, pretreated with Torin, as indicated and then stimulated with matched patient EPSC- or hGIC-CM. ($n = 3$; mean $\pm$ SEM, two-way ANOVA Tukey test).

Data information: **$P \leq 0.01$, ***$P \leq 0.001$.

Heidelberg classifier (Capper *et al*, 2018) (Table EV4). Patient-matched fibroblasts derived from the dura mater of the same patients were reprogrammed to EPSC (Yang *et al*, 2017, 2019), and iMGL was generated according to published protocols (Muffat *et al*, 2016; Fig EV5B). To confirm the progression of the differentiation process, cells were analysed by flow cytometry at different stages for markers of stemness, early yolk sac myelogenesis and mature microglia (Muffat *et al*, 2016; Abud *et al*, 2017; Douvaras *et al*, 2017; Pandya *et al*, 2017). Expression of Sox2 and NANOG was confirmed in EPSC, while all other markers were negative (Fig 7F, and Fig EV5C). Yolk sac embryoid bodies (YS-EBs) no longer expressed these stemness markers while gaining expression of CD41, c-kit, CD235a and PU.1, previously described as expressed before definitive haematopoiesis or the establishment of embryonic circulation (< E8.5) (Figs 7F and EV5C). Markers specific to microglia, P2RY12, Iba1 and TMEM119 were upregulated in mature iMGL (Figs 7F and EV5B and C), and complete differentiation was confirmed by immunocytochemistry for PU.1 and CD11b (Fig 7G) and double-labelling immunofluorescence for Iba1 and TREM2 (Fig 7H). Moreover, CD49d expression, specific to monocyte-derived macrophage, was not detected in the differentiated iMGL (Figs 7F and EV5C).

Next, hGIC-conditioned media was applied to the syngeneic EPSC-iMGL. Conditioned media from the syngeneic EPSC (from which the iMGL were differentiated; EPSC-CM) was used as a control. Similar to our findings in the mouse setting, p-S6 (S240/244) was upregulated by hGIC-secreted factors (Figs 7I and EV5D). Furthermore, increased p-STAT3 levels were detected in iMGL upon treatment with hGIC-CM and the effect was mTOR-dependent, as shown by p-STAT3 levels returning to basal levels upon Torin treatment (Figs 7J and EV5D). A trend towards an increase in p-NF-κB (p-P65) was also observed under hGIC-CM treatment, although not significant. Importantly, levels of p-NF-κB (p-P65) were significantly increased under hGIC-CM and Torin treatment (Figs 7K and EV5D). Treatment with EPSC-CM had no effect on the phosphorylation levels of these markers.

To validate in a human *in vitro* setting that this phenotype is microglia-specific and not observed in peripheral macrophages, we analysed mTOR signalling in tumour-conditioned peripheral blood monocytes/macrophages, obtained from healthy donor (Appendix Fig S3A). The phosphorylation levels of S6 (Appendix Fig S3B), STAT3 (Appendix Fig S3C) and NF-κB (Appendix Fig S3D) remained unchanged in CD49d$^+$ human monocytes/macrophages treated with EPSC-CM or hGIC-CM, validating the microglia-specific effect of GIC-secreted factors on mTOR signalling.

We show in a novel syngeneic *in vitro* model that iMGL conditioned with hGIC-CM upregulate mTOR and that this impacts STAT3 and NF-κB activity in a similar fashion to that found in murine models.

## Discussion

We show that GIC induce mTOR signalling in TAM-MG but not TAM-BMDM in *in vivo* and *in vitro* mouse models of GBM as well as in a human GIC/iMGL *in vitro* assay. The mTOR-dependent regulation of STAT3 and NF-κB activity promotes an immunosuppressed phenotype in TAM-MG, which hinders effector T-cell proliferation and immune reactivity and contributes to tumour immune evasion.

We describe increased mTOR signalling in TAM-MG but not in TAM-BMDM in mouse models of GBM. mTOR signalling positively correlates with TAM-MG enrichment in human GBM samples but not with TAM-BMDM at the transcriptomic level, supporting the translational relevance of our findings in mouse models. Our data imply that ontogeny affects the way microglia and monocyte-derived macrophages respond to GIC-secreted factors, which extends previous work showing that the transcriptomic profiles of TAM-MG and TAM-BMDM differ when exposed to the same tumour microenvironment (Bowman *et al*, 2016; Muller *et al*, 2017). Recent reports have identified mixed transcriptional states in the TAM population in glioma patients (Szulzewsky *et al*, 2015, 2016; Gabrusiewicz *et al*, 2016), and here, we identify mTOR signalling as a key driver of this intratumoral TAM heterogeneity.

We demonstrate that the pro-tumourigenic role of TAM-MG in GBM is mediated by mTOR, as reduced tumour growth and increased survival were observed upon genetic silencing of the pathway in these cells in GL261 allografts. Although a contribution to the observed phenotype by the small proportion of TAM-BMDM also targeted by the *Cx3cr1*-Cre driver cannot be entirely excluded, it seems unlikely as both the transcriptomic analysis in the *in vivo* model and the signalling analysis in the *in vitro* model highlight the lack of significant mTOR activity in TAM-BMDM. The transcriptomic profile of *Cx3cr1-Rheb1*$^{\Delta/\Delta}$ tumours demonstrated a shift in the immune landscape with an overall decrease in the negative regulation of T cells in the TME and a change in T-cell state from an exhausted to an active profile, suggesting a capacity to mount an anti-tumour adaptive immune response. This is further demonstrated by a change in the immune composition of *Cx3cr1-Rheb1*$^{\Delta/\Delta}$ tumours, defined by reduced numbers of microglia, while immune cells that have infiltrated from the peripheral circulation are more numerous, including effector T cells and TAM-BMDM. Transcriptomic analysis of *Rheb1*$^{\Delta/\Delta}$ GL261 TAM-MG revealed a re-education of these cells to an immune reactive and anti-tumour profile, with an enrichment for pathways linked to the regulation of Th1, Th2 and IFN signalling as well as pathways linked to recruitment, proliferation and priming of APC and cytokine signalling pathways. The predicted impact of this transcriptional deregulation is an increase in the stimulation of the adaptive immune system by innate immune cells and consequently an increase in effector and cytotoxic T cells within the tumour; an effect which was confirmed by tissue and flow cytometry analyses, which revealed an increase in CD4$^+$ and CD8$^+$ T cells, and no significant changes in FoxP3$^+$ cells in the *Cx3cr1-Rheb1*$^{\Delta/\Delta}$ TME as well as an increase in infiltration, proliferation and effector function of CD4$^+$ and CD8$^+$ T cells.

Glioblastoma multiforme are lymphocyte depleted with a high infiltration of TAM (Mirzaei *et al*, 2017; Thorsson *et al*, 2018; Woroniecka *et al*, 2018). Within the tumour-infiltrating lymphocytes, Tregs are the most numerous population and can suppress T helper cell and CTL responses (El Andaloussi and Lesniak, 2006; Mirzaei *et al*, 2017), while CD8$^+$ and CD4$^+$ T cells are exhausted (Thorsson *et al*, 2018; Woroniecka *et al*, 2018). Moreover, CD8$^+$ and CD4$^+$ T cells which do infiltrate the tumour seem unable to mount an anti-tumour effector response in GBM (Learn *et al*, 2006). T-cell exhaustion is known to result from an excessive and continuous stimulation by APC and cytokines, resulting in sustained expression of

inhibitory receptors and the lack of a productive anti-tumour effector response (Mirzaei et al, 2017). TAM contribute to T-cell dysfunction in GBM via their immunosuppressed phenotype, characterised by reduced expression of pro-inflammatory factors, antigen-presenting machinery and T-cell activation factors (Poon et al, 2017). In our Cx3cr1-Rheb1$^{\Delta/\Delta}$ model, T-cell effector profiles were stimulated, as shown by the increased proliferation and expression of cytotoxic factors such as IFN$\gamma$, granzyme b and perforin, in keeping with a scenario where mTOR significantly contributes to TAM-mediated T-cell dysfunction in GBM. Moreover, a distinct deregulation of cytokine signalling pathways was identified in our Cx3cr1-Rheb1$^{\Delta/\Delta}$ model, most notably those regulated by STAT3 (anti-inflammatory cytokines) and NF-$\kappa$B (pro-inflammatory cytokines). The survival benefits of STAT3 inhibition have been shown in a GL261 model, where the expression of cytokines promoting tumour growth, such as IL-10 and IL-6, was blocked (Zhang et al, 2009). Further work by Hussain et al (2007) illustrates the potential effect of these STAT3-regulated cytokines, expressed by TAM, on the proliferation of effector T cells and TCR-mediated signalling. Importantly, STAT3 is upregulated in TAM in human GBM and considered an attractive therapeutic candidate (Heimberger and Sampson, 2011; Wei, Gabrusiewicz and Heimberger, 2013; Chang et al, 2017; Poon et al, 2017). We show here that mTOR signalling increases STAT3 activity and inhibits NF-$\kappa$B in TAM-MG in different GBM models, therefore hampering APC immune reactivity as well as effector T-cell proliferation and immune response via the expression of anti-inflammatory cytokines.

In our study, we have taken advantage of methodologies to derive induced microglia (iMGL) from EPSC (Muffat et al, 2016) to develop a new in vitro assay. GIC were established from human GBM, and hGIC-CM obtained therefrom were incubated with the syngeneic iMGL to assess the relevance of the results of our mouse models in humans on a patient-specific basis. mTOR signalling positively correlated with TAM-MG enrichment but not with TAM-BMDM at transcriptomic level in the TCGA samples, a finding that was most prevalent in mesenchymal tumours, and thus, we applied the assay to hGIC/iMGL derived from GBM classified as belonging to the mesenchymal subtype. We reason that as no significant differences in the levels of TAM-MG were previously observed across the molecular subgroups (Bowman et al, 2016), the strong correlation we observed in mesenchymal GBM was not due to a higher number of TAM-MG in this subgroup. We showed that hGIC derived from mesenchymal GBM triggered activation of mTOR signalling in iMGL and mTOR-dependent regulation of STAT3 and inhibition of NF-$\kappa$B signalling, in line with our findings in mouse models. Interestingly, a key characteristic of the mesenchymal subgroup of GBM is its strong association with immune-related genes and an enrichment of infiltrating immune cells (Chen & Hambardzumyan, 2018; Behnan et al, 2019), thereby raising the possibility that mesenchymal-specific features of the hGIC phenotype might be responsible for inducing mTOR signalling in TAM-MG.

We demonstrate that GIC-secreted factors are sufficient to increase mTOR activity in microglia, although this does not exclude the possibility that factors secreted by other cells, including non-GIC tumour cells, may contribute to this phenotype. The secretome of GIC has been characterised (Formolo et al, 2011; Polisetty et al, 2011), but only little is known on the functional impact

of specific secreted factors on TAM phenotypes. We show here that GIC-CM contains factors capable of inducing mTOR pathway activation in TAM-MG in both humans and mice. A study comparing conditioned media of GIC and healthy NSC identified several inflammatory and growth factors, including potential mTOR stimuli (Okawa et al, 2017). While it is likely that a combination of these factors is responsible for the phenotype, osteopontin and lactate emerge as strong candidates (Lamour et al, 2010; Okawa et al, 2017). Osteopontin acts via several integrins known to influence PI3K/AKT/mTOR signalling (Ahmed & Kundu, 2010). It regulates migration, phagocytosis and the expression of inflammatory factors in microglia (Yu et al, 2017), including in a GBM context (Ellert-Miklaszewska et al, 2016; Wei et al, 2019). Osteopontin expression correlates with poorer survival in GBM (Atai et al, 2011; Wei et al, 2019), and its expression is enriched in mesenchymal as compared to classical and pro-neural tumours (Wei et al, 2019). Strikingly, remarkably similar findings to those seen in our mouse model were described in the TME of a GL261 allograft GBM model upon depletion of osteopontin (Wei et al, 2019). Lactate was also shown to be highly expressed by GIC and has been proposed as a prognostic marker for GBM (Marchiq & Pouyssegur, 2016). It contributes to acidification of the TME, which polarises TAM (Colegio et al, 2014; Romero-Garcia et al, 2016; Mu et al, 2018) therefore promoting immune evasion and tumour growth (Lui & Davis, 2018), possibly via interaction of lactate with the GPR65 receptor on TAM (Lailler et al, 2019). A study examining the effect of lactate and hypoxia on macrophages demonstrated an increase in mTOR signalling, which is suggested to be responsible for acquired M2-like phenotype with the inhibition of pro-inflammatory cytokines production (Zhao et al, 2019). It is therefore conceivable that differences in the threshold of lactate- and/or osteopontin-dependent mTOR activation in TAM-MG and TAM-BMDM may explain the different phenotype and function of these two populations in the TME.

Despite the importance of the deregulation of mTOR signalling in driving GBM growth, drugs aimed at targeting this pathway have so far failed in clinical trials (Jhanwar-Uniyal et al, 2019). Our results raise the possibility that tumour cells should not be the primary target of mTOR inhibition. Infiltration of CD8$^{+}$ and CD4$^{+}$ T cells but not FoxP3$^{+}$ Treg cells, as observed in our Cx3cr1-Rheb1$^{\Delta/\Delta}$ tumours, correlates with long-term survival in GBM patients (Heimberger et al, 2008; Yang et al, 2010; Abedalthagafi et al, 2018), hence providing the rationale for immunotherapies aimed at modifying the infiltration or immune reactivity of the T-cell population, such as drugs targeting inhibitory checkpoints. However, immune checkpoint inhibitors such as anti-CTLA-4 and anti-PD-1 antibodies have had little success as monotherapies in the treatment of GBM (Chen & Hambardzumyan, 2018), suggesting that blockade of immune checkpoints alone is not sufficient to restore anti-tumour immune functions in the GBM TME. Our observation that increased CD8$^{+}$ and CD4$^{+}$ tumour-infiltrating lymphocytes, induced by mTOR inhibition in Cx3cr1$^{+}$ TAM, correlates with reduced tumour growth supports further exploration of this approach and raises the possibility that precision targeting of the mTOR pathway, for example by nanoparticle-based drug delivery, the efficacy of which have already been demonstrated in liver and breast cancer (Huang et al, 2012; Singh et al, 2017), could be a viable approach in combination with existing T cell-targeted immunotherapies to condition the TME

towards a pro-inflammatory state, which is potently anti-tumourigenic.

# Materials and Methods

### In vitro cell culture

#### Mouse cell cultures

mNSC were previously derived from the subventricular zone of $Pten^{fl/fl};p53^{fl/fl}$ C57BL/6 mice. The $Pten^{-/-};p53^{-/-}$ mGIC were previously generated by *in vitro* recombination of these $Pten^{fl/fl};p53^{fl/fl}$ mNSC using adenovirus expressing Cre recombinase and confirmed as GIC following intracranial injection of these cells into the striatum of non-recombined mice of similar genetic background, which led to the development of malignant tumours (Jacques *et al*, 2010). $Pten^{fl/fl};p53^{fl/fl}$ mNSC and $Pten^{-/-};p53^{-/-}$ mGIC were maintained in culture as previously described, in DMEM/F12 medium (Gibco, 31330-038), supplemented with growth factors (B27 (2%, Gibco, 12587-010), mEGF (20 ng/ml, Preprotech, 315-09) and hFGF (20 ng/ml, Preprotech, AF-100-18B)). Culture plates were coated with poly-L-lysine (Sigma, P1524; 0.01 mg/ml for 30 min at room temperature-RT) and laminin (Sigma, L2020; 1:160 for 30 min at 37°C) (Zheng *et al*, 2008). The GL261 murine glioma cells were a gift from Dr Jeffrey E. Segall in 2018 (Albert Einstein College of Medicine, USA). The cells were cultured in DMEM supplemented with 10% heat-inactivated foetal bovine serum, penicillin, streptomycin and glutamine. Conditioned media was collected after 24 h from 80% confluent cultures, filtered through 0.2 μm filter and stored at −80°C. BMDM were differentiated and primary microglia isolated using standard protocols (Weischenfeldt & Porse, 2008; Chen *et al*, 2013). Once established, BMDM and microglia were cultured in the same DMEM media supplemented with 1% penicillin–streptomycin solution.

#### Human fibroblasts cultures from dura mater, establishment and maintenance

Patient consent and ethical approval were available for the study (08/H0716/16 Amendment 1 17/10/2014). Thin strips of dura mater were sliced and triturated with a scalpel. The tissue was then digested with trypsin for 5 min at 37°C. The reaction was stopped with culture media (DMEM, Glutamax, 10% foetal calf serum, 2% L-glutamine and 1% penicillin–streptomycin). Samples were centrifuged and resuspended in fresh media and plated in 6-well plates (Corning #BC010). Media was topped up 1 week later and then changed every 48 h. RNA and DNA were extracted from cell pellets using the RNA/DNA/Protein Purification Plus kit (NORGEN, #47700), following the manufacturer's protocol.

#### EPSC generation from fibroblasts cultures

Fibroblasts reprogramming was performed following a previously described protocol (Yang *et al*, 2017, 2019). Briefly, the fibroblasts isolated from dura mater were mixed with episomal vectors expressing Oct4, c-Myc, Klf4, Sox2 (OCKS 4F, 5 μg) and Rarg, Lfh1 (RL 2F, 5.0 μg), electroporated with Amaxa Nucleofector (Lonza, Germany) and then plated on SNL feeders with M15 media (Knockout DMEM (Invitrogen), 15% foetal bovine serum (Hyclone), 1X glutamine–penicillin–streptomycin (Invitrogen), 1X non-essential amino acids

(Invitrogen)). Upon the appearance of colonies, media was replaced with EPSCM (DMEM/F12 (Invitrogen), 20% knockout serum replacement (Invitrogen), 1X glutamine–penicillin–streptomycin, 1X non-essential amino acids (Invitrogen), 0.1 mM β-mercaptoethanol (Sigma), 106 U/ml hLIF (Millipore) supplemented with the following inhibitors: CHI99021 Tocris 1 μM, JNK Inhibitor VIII Tocris 4 μM, SB203580 Tocris 10 μM, A-419259 Santa Cruz 1 μM and XAV939 Stratech 1 μM). Colonies were isolated and plated in 24-well SNL feeders' plates for expansion and characterisation. RNA and DNA were extracted from cell pellets using the RNA/DNA/Protein Purification Plus kit (NORGEN, #47700), following the manufacturer's protocol.

#### GIC isolation

Patient consent and ethical approval were available for the study (08/H0716/16 Amendment 1 17/10/2014). Fresh GBM tissue was sliced and triturated with razor blade, dissociated with Accumax (sigma, A7089) at 37°C for 10 min and then filtered through a 70 μm cell strainer. Dissociated cells were plated on laminin-coated 6-well plate in NeuroCult NS-A Proliferation kit media (STEMCELL, 05751), heparin (2 μg/ml; Gibco 12587-010), mEGF (20 ng/ml, Preprotech, 315-09) and hFGF (10 ng/ml; Preprotech, AF-100-18B). Established cells were passaged when 70% confluent, frozen in Stem Cell Banker (Ambsio ZENOAQ, 11890) and stored in liquid nitrogen. RNA and DNA were extracted from cell pellets using the RNA/DNA/Protein Purification Plus kit (NORGEN, #47700), following the manufacturer's protocol. Cell lines used in this study are primary lines either derived from mouse brains or human tumours, and they have been characterised by transcriptomic profiling and cultured according to current practice, including contamination screening.

#### EPSC differentiation to iMGL

iMGL cells were generated following Muffat *et al* published protocol (Muffat *et al*, 2016), with some changes including culturing YS-EBs in oxygen-deprived environment (5% $O_2$) as previously described (Abud *et al*, 2017).

#### In vitro stimulation of tumour-infiltrating T cells

Following tumour tissues processing, single-cell suspensions were plated in 24 well plates with DMEM supplemented with 1% of penicillin and streptomycin. The cultures were stimulated with cell stimulation cocktail (eBioscience, 00-4970) composed of PMA and ionomycin, in combination with the protein transport inhibitor cocktail (eBioscience, 00-4980). Following overnight incubation, the cells were collected and stained according to the intracellular flow cytometry protocol.

#### Isolation of human blood-derived monocytes/macrophages

Peripheral blood was collected from healthy volunteers. Monocytes/macrophages were purified from the samples using the RosetteSep Human Monocyte Enrichment Cocktail Kit (StemCell, 15028) and Lymphoprep density gradient medium (StemCell, 07801) following the manufacturer protocol. The highly enriched monocyte population was then cultured using RPMI culture medium and conditioned with conditioned media where specified in figure legends. Conditioned cells were then process by flow cytometry. CD49d expression was assessed for purity of the population.

### Analysis of signal transduction events

A total of $1 \times 10^6$ mouse BMDMs, mouse primary microglia and iMGL per condition were treated and stimulated as indicated. Cells were pretreated for 60 min with 100 nM rapamycin (Calbiochem, 552310), 500 mM Torin (Tocris, 4247) or 10 μM LY294002 and then stimulated with 100 ng/ml of LPS from E. Coli 0111:B4 (Sigma, L4391) or with mNSC-CM, mGIC$^{Pten-/-;p53-/-}$-CM and GL261-CM. Protein analysis was carried out by Western blot or flow cytometry as described in supplemental experimental procedures. Cytokines were analysed at RNA level. Samples for RNA analysis were extracted and processed following manufacturer's protocol, using the Qiagen RNA extraction microkit (Qiagen, 74004) and KiCqStart SYBR Green Primers (Sigma).

### In vivo

### Mice

All procedures were carried out according to the Home Office Guidelines (Animals Scientific Procedures Act 1986, PPL 70/6452 and P78B6C064). Mouse models have been previously reported (Zou et al, 2011; Yona et al, 2013; Bowman et al, 2016). The Cx3cr1$^{Tm2.1}$Cre$^{ERT2}$ mice, developed in Steffen Jung's lab (Yona et al, 2013), were purchased from Jackson Lab (#021160) and bread with Rheb1$^{fl/fl}$ mice, shared with us by Paul Worley, Johns Hopkins Department of Neuroscience (Zou et al, 2011). Tamoxifen (T5648, Sigma) was diluted in 100% ethanol at 37°C, then in sunflower seed oil (Sigma, S5007 - 10 mg/ml) and injected i.p. 3 weeks before tumour initiation as previously described (Bowman et al, 2016). Two intraperitoneal injections of 1 mg of tamoxifen each were administrated within 48 h.

### Brain tumour models

Intracranial injections of GL261 cells were performed 3 weeks post-tamoxifen injections as previously described (Bowman et al, 2016). Briefly, the animals were anesthetised using isoflurane and analgesic was injected prior to surgery. Using a stereotactic frame, cells were injected into the right frontal cortex (2 mm posterior and 2 mm lateral from the bregma anteriorly and the lambda at a depth of 2–3 mm). GL261 cells were injected at a concentration of $2 \times 10^4$ cells/5 μl in PBS at 6 weeks of age. The animals were scarified around 3 weeks post-tumour initiation.

### MRI and quantitative assessment of tumour volume

Tumour imaging was carried out on Bruker ICON 1T preclinical MRI system using a Paravision software. Before the imaging experiments, mice were anaesthetised with isoflurane/$O_2$ [4% (v/v)] and maintained on isoflurane/$O_2$ [2% (v/v)] throughout the experiment. T2_rare scans were carried out with the following parameters: 2,505 ms repetition time; 85 ms Echo time; 21.25 ms Echo spacing; 11 slices 0.8 mm thick each; $89 \times 89$ voxels image size; $0.191 \times 0.191$ mm resolution; and $17 \times 17$ mm field view, 16 averages. Images were analysed on the VivoQuant software, by manually defining tumour region with 3D ROI tool.

### Tumour tissue processing

For all tissue analyses, mice were anaesthetised with Euthatal and transcardially perfused with PBS. Tumour tissues were isolated by macrodissection and dissociated using the Brain Tumour Dissociation Kit (BTDK; Miltenyi, 130-095-942) and a single-cell suspension generated using the gentleMACS™ Dissociator. All single-cell suspensions were filtered through a 40 μm mesh filter. Normal brain and brain tumour tissues were incubated with Myelin Removal Beads (Miltenyi, 130-096-733). Cells were then counted and resuspended in flow cytometry buffer. Alternatively, tissues were collected following perfusion and incubated in 4% PFA for 2 days prior paraffin embedding or processed in 30% glucose overnight prior to freezing in OCT. Antibody labelling was carried out using standard protocols.

### Immunoassay techniques

### Western blotting

For Western blot, lysate preparation and analysis was performed as previously described (Weichhart et al, 2008; Badodi et al, 2017). Proteins were extracted with RIPA buffer (Santa Cruz, sc 24948A) on ice for 45 min with frequent vortex. Equal amounts of proteins were separated by SDS–PAGE (NUPAGE 4-12% bisacrylamide NP0335) and incubated with primary antibody. Vinculin was used as housekeeping protein for loading control. Results were visualised using a BIORAD ChemiDoc MP Imaging. Protein quantification was performed using Fiji image analysis software.

### Flow cytometry

Following tumour tissues processing, single-cell suspensions were resuspended in FC blocked (CD16/32, BD #553141) on ice and then incubated with antibody master mix. After washing, the cells were stained with a fixable viability dye-e506 (eBioscience) in PBS. For extracellular staining only, cells were fixed in 2% PFA and resuspended in flow cytometry buffer. For intracellular staining, cells were resuspended in Foxp3 Intracellular staining kit (eBioscience) according to the manufacturer's protocol. All flow cytometric analysis was performed using a BD LSRFortessa device, and sorting was performed on an Aria fusion, with a FACSDiva software version 8. Data were transferred and analysed using the FlowJo software V10 (Tree Star, Oregon, USA).

The gating strategy employed was as followed—cells were first selected based on size using the FSC-A and SSC-A. Then, doublets were removed from the analysis using SSC-A versus SSC-W and FSC-A versus FSC-H. Lastly, live cells were selected for by gating on cells negative for the fixable viability dye-e506. Gating for T cells employed CD45 and CD3 markers as well as CD4 and CD8 to define subpopulation. Gating for TAM population relied on CD11b$^+$ CD45$^{low}$ P2RY12$^+$ CD49d$^-$ marker expression for TAM-MG and CD11b$^+$ CD45$^{high}$ P2RY12$^-$ CD49d$^+$ for TAM-BMDM.

### Immunohistochemistry and tissue analysis

Frozen and FFPE tumour tissues were obtained for the GL261 allograft model and the genetic models Ntv-a;PDGFB+Shp53, $Pten^{-/-}$; $p53^{-/-}$, the $Pten^{-/-};p53^{-/-};Idh1^{R132H}$ and PDGFB (Jacques et al, 2010; Bowman et al, 2016; Zhang et al, 2019). The mouse samples were processed at UCL IQpath laboratory. Dewaxing, antigen retrieval and pretreatment with appropriate serum were performed as per published protocols (Badodi et al, 2017). One section per biological replicates was analysed and quantified for each mouse model, and the number of biological replicates per mouse model is indicated in the figure legends. Stained cells were cultured and

conditioned on coverslips. After incubation time, the cells were fixed and permeabilised with 0.1% triton. Following pretreatment with appropriate serum, cells were incubated with primary and then secondary antibody for 1 h each. Coverslips were mounted on slide using prolong gold anti-face mounting with DAPI (Life Technology, P36981). An isotype control was performed to validate the quenching of fluorescence from reporter gene. A Leica DM5000 Epi-Fluorescence microscope was used for analysis of immunocytochemistry and on Zeiss confocal 880 with Zen 2.3 SP1 program for immunohistochemistry. Definiens Tissue Studio Software (Definiens AG) was used for quantification.

### Antibody list

The following antibodies were used for immunoassays: p-S6 Ser240/244 (Cell Signalling, 5364 and 6520), S6 (Cell Signalling, 2217), p-4EBP1 Thr37/46 (Cell Signalling, 2855), 4EBP1 (Cell Signalling, 9644), p-AKT Ser473 and Thr308 (Cell Signalling, 9271 and 9275), AKT (Cell Signalling, 4691), p-STAT3 Tyr705 (Cell Signalling, 4113 and BioLegend, 651021), STAT3 (Cell Signalling, 9139), p-P65 (Cell Signalling, 3031 and 5733), P65 (Cell Signalling, 8242), Vinculin (Sigma, V4505), Iba1 (Wako, 019-19741), CD11b (eBiosciences, 17-0112-82) and (Abcam, ab8878), CD45 (BioLegend, 103105), Ki67 (BioLegend, 652413), Granzyme b (BioLegend, 337221) and Perforin (BioLegend, 154406). IFNγ (BioLegend, 505826), CD49d (BioLegend, 103618 and 304313), CD44 (BioLegend, 10349), CD62L (BioLegend, 104438), CD3 (BioLegend, 152303), CD4 (BioLegend, 100427), CD8 (BioLegend, 100722 and 100732), PU.1 (Cell Signalling, 2258), c-kit (Abcam, ab212518), CD41 (BioLegend, 303702), CD235a (Life Technology, 14-9987-82), P2RY12 (Atlas, HPA014518 and 848006), TREM2 (Abcam, ab86491),TMEM119 (Sigma, HPA051870), MHCII (BioLegend, 107607), F480 (BioLegend, 123130), Sox2 (Santa cruz, sc-17320) and NANOG (BioLegend 16H3A38).

### Quantification of immunocytochemistry and immunohistochemistry

For co-expression analysis on tissue, a protocol was developed on Definiens software to automatically identify DAPI and the markers of interest. The protocol was composed of the following steps: tissue detection, ROI detection and cellular analysis. In tissue detection, the programme is taught to recognise tissue versus the glass slide. Under ROI detection, the areas of interest were defined manually. Under cellular analysis, DAPI or haematoxylin stain was used to identify nucleated cells and the cell marker was used to define area of nucleated cells. The software was trained on a subset of tissue regions and then automatically applied to entire slides. The number of cells co-expressing the marker of interest and the number of DAB-positive cells was then calculated. For co-expression analysis on stained cells, the colocolisation threshold tool on ImageJ was used to calculate the % of voxels colocalised between channels.

### In silico analysis

### Processing and analysis of RNA-Seq and DNA methylation data

Total RNA was extracted from sorted cells, cultured cells and FFPE tissue. TAM-MG were sorted by gating on single live CD11b$^+$ CD45$^{low}$ cells. Cells were sorted into TRIzol (Sigma, #93289), and using chloroform (0.2 ml per 1 ml of TRIzol), the aqueous phase

was separated by centrifugation, from which RNA was extracted using the RNA extraction Microkit (Qiagen, 74004) following manufacturer's protocol. Human GBM and GL261 bulk tumour RNA was extracted from tumour tissue scrapped off FFPE slides, using the FFPE RNA/DNA Purification Plus Kit (NORGEN, 54300) following manufacturer's protocol. RNA and DNA were extracted from cultured cells using the RNA/DNA/Protein Purification Kit (NORGEN, 47700) following manufacturer's protocol.

Library preparation (SmartSeq2 and NexteraXT) and sequencing (HiSeq4000 75 bp pair end (Illumina) were carried out at Oxford Genomics Centre. RNA-Seq data were processed using two separate pipelines. For hierarchical clustering and principal component analysis, transcript-level expression was estimated directly using the pseudoalignment package Salmon (Patro *et al*, 2017). The results were then aggregated to obtain gene-level expression estimates in units of transcripts per million (TPM), which are normalised for library size and transcript length. For the purpose of identifying deregulated genes, gene counts were first estimated using the gapped alignment software STAR (Dobin *et al*, 2013). The reference genomes used in both pipelines are Ensembl GRCm38 (release 90) for mouse data and Ensembl GRCh38 (release 90) for human data. DE genes are computed using the R package edgeR (Robinson *et al*, 2010), requiring a minimum absolute fold change of 1.5 and a false discovery rate lower than 0.05. Functional analysis of DE gene lists was carried out on the IPA software.

DNA methylation data for GIC lines were assayed on the Illumina Human Methylation EPIC or 450K microarrays. EPIC array data were reduced to include only the same probes as the 450K data (these have the same chemical design and are therefore comparable). Raw array data were first preprocessed using the ChAMP package in R to remove failed detections and probes with known design flaws (Feber *et al*, 2014), before normalisation using the SWAN algorithm (Maksimovic *et al*, 2012).

### Published transcriptomic dataset used in this study

Preprocessed gene expression data from the TCGA-GBM RNA-Seq and the Gravendeel-GBM Microarray repository were downloaded from the GlioVis data portal (Bowman *et al*, 2017), resulting in 138 samples from the TCGA dataset and 135 samples from the Gravendeel dataset. RNA-Seq data published by Bowman *et al* (2016) of TAM-MG, TAM-BMDM, healthy microglia and healthy monocytes from the GL261 model were obtained from the GEO database (GSE86572).

### Computing GBM subgroup based on DNA methylation and transcriptomic profile

Subgroups of GBM have previously been described based on analysis of methylome data (Sturm *et al*, 2012). Our bulk FFPE and GIC samples were assigned to subgroups on this basis using a published random forest classifier (Capper *et al*, 2018), which requires raw methylation array data as an input. Using transcriptomic data from the bulk FFPE and GIC samples as well as data from the TCGA cohort, the subgroup were computed according to Wang *et al* (2017) using the provided software tool.

### Evaluation of immune cell infiltration

Tumour-infiltrating immune cells were calculated using the CIBER-SORT algorithm (Newman *et al*, 2015). CIBERSORT is an analytical

tool, with a gene expression signature of 547 marker genes, used for quantifying the infiltrated immune cell composition fractions. LM22 is the annotated gene signature matrix defining 22 immune cell subtypes, including seven types of T cells, naïve B cells, memory B cells, plasma cells, resting and activated NK cells, monocytes, M0-M2 macrophages, resting and activated dendritic cells, resting and activated mast cells, eosinophils and neutrophils, which were downloaded from the CIBERSORT web portal (http://cibersort.stanford.edu/). Immune cell fraction of all the bulk RNA-Seq samples was analysed using the CIBERSORT algorithm.

### Calculation of gene signature enrichment scores with ssGSEA

Gene signatures that discriminate TAM-MG and TAM-BMDM in the context of a mouse tumour model were obtained from a study by Bowman *et al* (2016). These were converted into human orthologs using the HomoloGene database provided by NCBI (https://www.ncbi.nlm.nih.gov/homologene, build 68), which resulted in human signatures of 340 genes for TAM-MG (from 378 mouse genes) and 377 for TAM-BMDM (from 458) (Table EV1). A human mTOR signature was obtained from the Molecular Signatures Database (mSIGDB, (Subramanian *et al*, 2005) (Table EV1). The NF-κB (hsa04064) was obtained from the KEGG database, the "negative regulation of lymphocyte" and "T-cell differentiation" signature were obtained from Luoto *et al* (2018), and the T-cell chemotaxis, antigen presentation, STAT3 and IFNγ signatures were obtained from the ssGSEA database (Table EV1).

The function gsva in the homonym package from the R Bioconductor repository was used to compute ssGSEA scores, with "method=ssgsea" and default settings. The ssGSEA scores were calculated for each of the signatures (Barbie *et al*, 2009), in the 148 TCGA-GBM samples as well as the $Rheb1^{fl/fl}$ and $Rheb1^{\Delta/\Delta}$ bulk GL261 tumours and TAM-MG RNA-Seq samples. Scores were standardised using a z transform across samples for each signature.

### Pearson correlation analysis

For the bulk samples, a correlation analysis between CIBERSORT immune cell fraction values and ssGSEA enrichment scores was performed separately for each group (fl/fl and Δ/Δ) in R, using the CRAN package Hmisc and functions rcorr with default settings, to extract Pearson coefficients. To compare correlation coefficients between the two sample groups, the Pearson scores were subtracted $(\Delta/\Delta - fl/fl)$ to obtain a differential correlation score.

Similarly, for the TAM-MG RNA-Seq samples, Pearson correlation was calculated across signatures ssGSEA enrichment scores. With the TCGA-GBM samples, the Pearson correlation was computed between mTOR signature scores and both TAM-MG and TAM-BMDM separately.

### Statistical analysis and data visualisation

All statistical analysis was completed using python V2.7 or GraphPad Prism Pro V8. Flow cytometry plots and histogram were plotted in Flow Jo V10. All other scatter plots, boxplots were plotted with python V2.7, R and relevant Bioconductor packages or GraphPad Prism Pro V8 and IPA software (QIAGEN Inc., https://www.qiagenbioinformatics.com/products/ingenuity-pathway-analysis).

Parametric data are presented as mean ± standard error of the mean (SEM). *P* < 0.05 was considered statistically significant, with *P* < 0.05, < 0.01 and < 0.001 represented with \*, \*\*, \*\*\*,

respectively. Further information on statistical analysis of specific datasets is indicated in the figure legends.

Schematics were created using bioRENDER (https://biorender.com/) or IPA software.

## Data availability

- RNA-Seq data: Gene Expression of the GL261 TAM-MG and the Bulk GL261 from $Cx3cr1$-$Rheb1^{-/-}$ and $Rheb1^{fl/fl}$ samples (Omnibus GSE147329 https://www.ncbi.nlm.nih.gov/geo/query/acc.cgi?acc = GSE147329)

Other datasets generated in this study are available at the following databases:

- RNA-Seq data: Gene Expression TCGA-GBM RNA-Seq normalised count reads, downloaded from http://gliovis.bioinfo.cnio.es/
- RNA-Seq data: Gene Expression TCGA-Gravendeel RNA-Seq normalised count reads, downloaded from http://gliovis.bioinfo.cnio.es/
- RNA-Seq data: Gene Expression of Bowman *et al* dataset Omnibus GSE86572 (https://www.ncbi.nlm.nih.gov/geo/query/acc.cgi?acc = GSE86572)
- Methylation Array: Omnibus GSE147329 (http://www.ncbi.nlm.nih.gov/geo/query/acc.cgi?acc = GSE147329)

**Expanded View** for this article is available online.

### Acknowledgements

This work is funded by grants from Brain Tumour Research (Centre of Excellence award to SM), The Willoughby Fund Trustees (studentship to AAD), Barts Charity (project and programme grants to SM), The Brain Tumour Charity (GN-000389 clinical research training fellowship to TM) and the Swiss Cancer League (JAJ). Part of the study was funded by the National Institute for Health Research to UCLH Biomedical research centre (BRC399/NS/RB/101410). SB is also supported by the Department of Health's NIHR Biomedical Research Centre's funding scheme. We acknowledge the use of data generated by the TCGA Research Network: https://www.cancer.gov/tcga. We thank Anoek Zomer and Joanna Kowal, Joyce lab, Ludwig Institute for Cancer Research, University of Lausanne, Switzerland, for their help in the design of the *in vivo* experiments. We thank Frances Balkwill, Barts Cancer Institute, QMUL for critically reading this paper.

### Author contributions

AAD and SM conceived the study, designed and interpreted experiments. AAD performed experiments and analysed results. NP and GR performed all computational analyses. LG and CV established the patient-derived cell lines (GIC and EPSC). TOM assisted with intracranial injection experiments. JR and NA identified and consented the patients for the study. SB obtained ethical approval and supervised human sample collection and pathological analysis. JAJ, RLB, DS, JW, AM, and ABH shared datasets and essential expertise. SM supervised the study. AAD, GR and SM wrote the manuscript with contribution from all authors.

### Conflict of interest

The authors declare that they have no conflict of interest.

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
