## [Review Process File · The EMBO Journal]

Microglia promote glioblastoma via mTOR-mediated immunosuppression of the tumour microenvironment.

Anaëlle Dumas, Nicola Pomella, Gabriel Rosser, Loredana Guglielmi, Claire Vinel, Thomas Millner, Jeremy Rees, Natasha Aley, Denise Sheer, Yun Wei, Anantha Marisetty, Amy Heimberger, Robert Bowman, Sebastian Brandner, Joanna Joyce, and Silvia Marino

DOI: [10.15252/embj.2019103790](https://doi.org/10.15252/embj.2019103790)

Corresponding author(s): *Silvia Marino (s.marino@qmul.ac.uk)*

Review Timeline:

Submission Date:	20th Oct 19
Editorial Decision:	21st Nov 19
Revision Received:	23rd Mar 20
Editorial Decision:	22nd Apr 20
Revision Received:	3rd May 20
Accepted:	8th May 20

Editor: *Daniel Klimmeck*

Transaction Report:

Dear Dr Marino,

Thank you for the submission of your manuscript (EMBOJ-2019-103790) to The EMBO Journal. Your manuscript has been sent to three reviewers, and we have received reports from all of them, which I enclose below.

As you will see, the referees acknowledge the potential interest and novelty of your results, although they also express a number of issues that will have to be addressed before they can support publication of your manuscript in The EMBO Journal. In more detail, the referees states that a number of concerns about additional experiments and controls required to establish more detailed characterisation of the TAM microglia populations involved (ref#1, ref#2) and additional mechanistic insights into the mTOR downstream signaling (ref#3). In addition, the reviewers raise a number of issues related controls required to exclude potential confounding factors, methods annotation, data representation, statistics, appropriate discussion oif the results and citation of literature references that would need to be conclusively addressed to achieve the level of robustness and clarity needed for The EMBO Journal.

I judge the comments of the referees to be generally reasonable and given their overall interest, we are happy to invite you to revise your manuscript experimentally to address the referees' comments.

Please let me know any time if you have additional questions or need further input on the referee comments.

Please see below for additional instructions for preparing your revised manuscript.

Thank you for the opportunity to consider your work for publication. I look forward to your revision.

Kind regards,

Daniel Klimmeck

Daniel Klimmeck, PhD
Editor
The EMBO Journal

- a point-by-point response to the referees' comments, with a detailed description of the changes made (as a word file).

- a word file of the manuscript text.

- individual production quality figure files (one file per figure)

- a complete author checklist, which you can download from our author guidelines (<https://www.embopress.org/page/journal/14602075/authorguide>).

- Expanded View files (replacing Supplementary Information)

Further information is available in our Guide For Authors:

The revision must be submitted online within 90 days; please click on the link below to submit the revision online before 19th Feb 2020.

Link Not Available

Referee #1:

In this manuscript, Dumas et. al. studied the role of GBM cells in modulating the immunosuppressive phenotype of tumor associated microglia (TAM-MG) via regulating the mTOR pathway involving STAT3 and NF- κ B activities. To study the role of the mTOR pathway in TAM-MG in the GBM microenvironment, the authors used multiple tools such as GBM allografts in genetically engineered mice in which mTORC1 signaling has been silenced in TAM-MG, as well as human iPSC-derived induced microglial-like cells and matched GBM cells. The authors proposed that microglia represent an unexpected target for mTOR inhibitors to treat brain tumors.

The experiments are conducted in a careful and thoughtful manner, and the data included in this manuscript are rich, and support most conclusions well. Overall, the report is strong and I would recommend acceptance following adequate responses to the following points:

Major points:

1. The authors state that the immunosuppressive phenotype of TAM-MG is induced by GBM-

initiating cells (GICs), however the study does not provide side by side comparison of the role of GIC and non-GIC on the phenomenon described. The conditions under which the experiments were conducted do not seem GIC specific, except if the authors have purified or isolated GICs in ways that are not described in the method section. Therefore, I would suggest that the authors revise throughout the manuscript the statements about the role of GICs.

2. Using the TCGA database the authors showed a positive correlation between mTOR pathway and TAM-MG in human GBM. How does this correlate with survival? Is this correlation also observed with other databases? Is this GBM specific or this is also seen in LGG?
3. In the conditioned medium experiments (Fig. 2), it is unclear if the medium in which the different cell types (mNSCs, mGIC) are the same. Not sure if microglia media was conditioned by the tumor cells or if conditioned tumor cell media was used to grow microglia and BMDM. Are microglia media and BMDM media the same? Also, results with non-conditioned media is not presented. If these points do not come across clearly, it makes the interpretations of the results difficult for the reader.
4. Fig. 2: It would be great to see WB with BMDM similarly to what is shown with microglia.
5. It would be important to demonstrate that the key findings of figure 2 are repeated in the second mouse model that the authors used for the data shown in figure 1 and 3 for example.
6. Fig 3: Absence of survival data reduces the impact about the results presented. The impact of these findings will be definitely strengthened with survival experiments demonstrating directly that disrupting mTOR pathway in microglia improves disease outcome.
7. Fig. 3D: Please indicate on the figure which marker(s) are presented (Iba1, pS6, DAPI?), specifically that GL261 cells seem to also be expressing GFP and that Cre+ cells are also YFP+.
8. Fig 5: The authors should make clear in the main text as well as on the figure which type of cells (TAM-MG WT or floxed for Rheb1) were used for these experiments. Also, without corresponding legend, it is hard to know and interpret.
9. Fig. 6J-L: Similar to what is presented in figure 2, it would be important to test a control conditioned medium (e.g. conditioned with human NSCs or astrocytes) and show that the increase of p-S6 for example in microglia is specific to GBM cells.
10. Fig. 6J-L: I would have liked to see some conditioned media experiments looking at expression level of these markers in human macrophages (derived from autologous iPSC, non-autologous blood derived primary macrophages or from monocytic leukemia lines that can be differentiated into macrophage-like cells for examples). If presented, these results would fully support the first statement of the discussion that GBM cells induce mTOR pathway specifically in TAM-MG and not in TAM-BMDM, both in murine and human models.
11. In the discussion, the authors suggest the role of Osteopontin in the phenomenon described in the study. One other potential candidate one can propose is lactate. Indeed, GBM cells are highly glycolytic and generate large amount of lactate as a result of their metabolic reprogramming supporting their proliferation. The effect of lactate on mTOR pathway has been described and could explain a lot of the results presented here and the difference between TAM-MG and TAM-BMDM may reflect a different threshold in lactate-dependent mTOR activation. The authors may want to acknowledge and discuss this.

Minor points:

1. Fig. 1A-C: It is unclear how many brains and how many sections per brain were analyzed and quantified. This should be included at least in the supplemental method section.
2. Fig. 2A: a drawing of a human brain is shown but the experiments were conducted with murine models. A drawing of a mouse brain may more appropriately illustrate the experiments.
3. Fig. 5 lettering should be revised as it does not correspond to the main text and the legend. Because of this, the section "mTOR-dependent expression of inflammatory cytokines in tumour-conditioned microglia promotes an anti-inflammatory phenotype via the regulation of STAT3 and

NF- κ B transcription factors" is difficult to follow.

4. Fig. 6 lettering shows some discrepancies with the main text and should be revised.

5. Page 16 first paragraph: The authors wrote: "Importantly, levels of p-NF- κ B were significantly increased under hGIC-CM and Torin treatment (Fig.6L)". Do they mean p-P65 as shown in the figure?

Referee #2:

Review: Dumas et al.

In this article, Dumas et al. aim to put forward the idea that glioma-derived factors induce mTOR signaling in tumor-associated microglia but not macrophage. The study benefits from utilizing both in vivo and in vitro models of glioma-immune crosstalk including syngeneic human iPSC-derived microglia culture. The authors also propose that tumor cells should not be the primary target for mTOR inhibition. This study addresses a hitherto less studied role of mTOR signaling in the innate compartments of brain immunity in the context of GBM brain tumors. However, the work is, at best, mainly descriptive and lack any insightful mechanistic or technical breakthroughs. Several critical points need to be addressed in order to make the findings more convincing. The authors have touched upon many of the clinically relevant aspects of GBM to introduce the study but have failed to reconcile them with experimental rigor. Hence, this article is not recommended for publication in its current form.

Major comments

-The initial section chooses to address the spatial differences in mTOR signaling in TAM compartments in various genetic syngeneic mouse models of glioma but does not explain the reason behind intra model differences. The IHC images showcase PDGFB-driven model where there is no difference between the tumor core or edge. The zoomed in version does not provide any additional. The subsequent use of GL261 model does not systematically take off and spatial heterogeneity remains unaddressed.

-The manuscript somewhat tries to contextualize a relatively dated transcriptomic classification scheme (Verhaak, 2010) with the current version but that does not add to the impact of the study. The detailed explanation, on the other hand, unnecessarily confuses the reader.

-In the light of differential overlap of Iba1+ and CX3Cr1+ microglial functions, the study should employ more functional markers for in vivo/ ex vivo phenotyping.

-Figure 2 data utilizes Pten-/- p53-/- GIC model to test the role of secreted factors. The study loses continuity from rather random choice of models for a particular set of experiments. Similar experiments need to be performed with at least another GIC model. It would be important to comment on the nature of secreted factors and their similarities/dissimilarities between different GIC models. Western blots for TAM-BMDM would better drive home the key finding of the study.

-Several figures lack important labels and information. Many of the supplemental figures find no mention in the manuscript. Figure 5 is missing two very important pieces of data mentioned in the manuscript.

-The authors fail to highlight the novelty of their iPSC MGL assay. The protocol is well established, and Figure 6 barely finds its way to utilize the model beyond standard characterization. The study does not comment on how it will specifically target TAM-MG mTOR signaling due any detailed novel cell-intrinsic mechanism. Geneset enrichment results are poorly explained.

Minor Comments

- Numerous typos; non-standard term usage (Figure S5 B)
- Figure 3 D does not have any color coding for markers.
- Figure S3 A is confusing regarding the validation of loss of Rheb.
- % of cell scale bars should include the gating strategy/ information for each graph.

Referee #3:

Dumas et al describe differences between the two main populations of Tumor Associated Macrophages (TAM): the bona fide infiltrating Macrophages (TAM-BMDM) and microglia (TAM-MG) in GBM. Combining data from in vitro systems including iPSC derived MG and GL261 allograft model, the authors describe mTOR pathway as an important regulator of TAM-MG but not TAM-BMDM. However, my main concern is that the mTOR downstream regulators that they propose need to be better characterized.

Major comments:

As mentioned above, my main concern is that the lack of evidence for the downstream regulators of mTOR, although STAT3 seems more convincing, there are not enough experimental evidence to conclude their statement "...concomitant reduction in expression of IL-12 mediated by the reduced phosphorylation of NF- κ B" (page 14, first paragraph). Torin treatment showing a modest increase on the phospho-NF- κ B is not enough to prove the connection of mTOR and NF- κ B in their specific context. Did they ever study the nuclear translocation of NF- κ B or any of the downstream targets of this transcription factor?

Moreover, throughout the analysis of their transcriptional data there is not enough data linking mTOR and NF- κ B or at least they didn't highlight it.

Regarding the gene expression data, it is also striking to me that they didn't show a dysregulation of mTOR signaling pathway in the Rheb1 KO system (Figure 5). That can be simply since Figure 5 is missing at least 3 panels (both in the text and figure legend there is A to G but in the figure, we only see till E with the derived inconsistency on the legend and text).

The authors base their conclusions on the transcriptional and functional differences between TAM-MG and TAM-BMDM. These differences are detected in vitro (Fig 2A) but less evident when they analyze TAM-MG and TAM-BMDM from tumors. In particular, the authors do not clearly explain the strategy followed to discriminate these cell types, and important point considering that they express similar cell-markers.

The authors should also consider and discuss whether the downregulation in TAM markers detected simply reflects decreased TAM numbers in tumors?

It is important to validate the quality of iPSC-derived microglia, which in similar setups has been shown to be closer to macrophages than microglia.

Additional comments:

Figs 2 C E are missing the data with the corresponding inhibitors. Also, as a general comment for Fig 2 C E G and I, it would be more clear if they state the inhibitor in the legend for each panel instead of making that general legend on the bottom.

Have the authors treated Cx3cr1:CreERT2 Rheb1 fl/fl with tamoxifen and waited a month before inducing GBM? This is an important point to assure that target genes are only deleted in MG and not macrophages.

Tables where not labeled so which table was which could be tracked.

Response to reviewers' comments

Referee #1:

In this manuscript, Dumas et. al. studied the role of GBM cells in modulating the immunosuppressive phenotype of tumor associated microglia (TAM-MG) via regulating the mTOR pathway involving STAT3 and NF-kB activities. To study the role of the mTOR pathway in TAM-MG in the GBM microenvironment, the authors used multiple tools such as GBM allografts in genetically engineered mice in which mTORC1 signaling has been silenced in TAM-MG, as well as human iPSC-derived induced microglial-like cells and matched GBM cells. The authors proposed that microglia represent an unexpected target for mTOR inhibitors to treat brain tumors. The experiments are conducted in a careful and thoughtful manner, and the data included in this manuscript are rich, and support most conclusions well. Overall, the report is strong and I would recommend acceptance following adequate responses to the following points:

Major points:

1. The authors state that the immunosuppressive phenotype of TAM-MG is induced by GBM-initiating cells (GICs), however the study does not provide side by side comparison of the role of GIC and non-GIC on the phenomenon described. The conditions under which the experiments were conducted do not seem GIC specific, except if the authors have purified or isolated GICs in ways that are not described in the method section. Therefore, I would suggest that the authors revise throughout the manuscript the statements about the role of GICs.

This is an important point which we are happy to clarify. All experiments presented in Fig2 and FigS2 were carried out on murine glioblastoma initiating cells (mGIC). These cells have been established and previously characterised by Zheng et al 2008 and Jacques et al 2010. These publications describe how neural stem cells (NSC) were isolated from the subventricular zone (SVZ) of mice bearing conditional alleles for *Pten* and *p53* (*Pten^{fl/fl}/p53^{fl/fl}*) and were recombined *in vitro* using an adenovirus expressing cre recombinase. Intra-cranial injection of these cells into the striatum of recipient mice of similar genetic background led to the development of high-grade gliomas. These data support the statement we make in this manuscript that the *Pten^{-/-}/p53^{-/-}* tumour cells are mGIC. We have now included more information on these cells in the M&M section "Mouse cell culture" and "Analysis of signal transduction events" (page 22).

We agree that a comparison between the effect of mGIC and non-mGIC on the microenvironment would be very interesting, but we do not feel this is currently technically feasible. Cancer stem cells (such as GIC) are defined by their stem-like properties of sustained self-renewal, extensive proliferative potential, and ability to initiate tumour formation following intracranial injection in mice. Other characteristics include their ability to differentiate into multiple lineage and the expression of stem cell markers. Availability of robust and defining markers would be essential to isolate GIC from non-GIC in the tumour bulk. However, whilst CSC/GIC markers such as CD133 do isolate cells with properties of GIC, they do not allow to exclude that cells not expressing CD133 at that moment in time are GIC. In fact, there is no absolute marker/s discriminating between GIC and non-GIC.

Therefore, while we feel we can claim that the mGIC secretome induces increase mTOR activity in TAM-MG, we should make it clearer that this does not imply that this is an exclusive property of GIC and non GIC may or may not have a similar effect. We have made changes in the results (page 7) and fifth paragraph of the discussion (page 20) to reflect this.

2. Using the TCGA database the authors showed a positive correlation between mTOR pathway and TAM-MG in

human GBM. How does this correlate with survival? Is this correlation also observed with other databases? Is this GBM specific or this is also seen in LGG?

These are interesting points which we have tried to address. We have compared survival between patient with the signature (high enrichment of mTOR and microglia) versus those without the signature. When looking at the survival over time between these two patient groups, a trend toward a worse survival is found for patients with the signature (orange) as compared to those without the signature (green; Fig.1A of this rebuttal). This is in line with the data obtained in mouse models. However, this pattern was not significant. Alternatively, we separated the samples based on expression of RPS6 and P2RY12 (those with high levels of both markers versus others). Similar survival patterns were observed with lower median survival in patients with high expression of RPS6 and P2RY12 (Fig.1B of this report). It is likely that the findings are not significant because of the small dataset (n numbers indicated on the survival graphs), which greatly limits the statistical power of the analysis. This analysis is further hindered by the aggressive nature of this disease, which leads to very short overall survival of patients.

In regard to whether the correlation exist in alternative datasets than the TCGA, we looked at additional datasets present on the Gliovis portal – we made use of the Gravendeel dataset. The Gravendeel dataset replicated our findings from the TCGA dataset: a significantly positive correlation between mTOR and TAM-MG enrichment but not between mTOR and TAM-BMDM enrichment scores was found. The data is discussed in the last results section and is displayed in fig.S7A (pages 15).

We have also checked whether the signature is present in the LGG tumours and used again the transcriptomic data of the TCGA. Astrocytomas display no significant correlation between mTOR and TAM-MG or TAM-BMDM signature enrichment. Oligodendroglioma instead do show a positive correlation between mTOR and TAM-MG although also between mTOR and TAM-BMDM, however both the slope (0.22) and r-square (<0.05) values are very low reflecting high variability between patients. Because of the lack of a strong correlation (see Fig.2 of this rebuttal) in LGG, the relatively low number of cases and the fact that we focus on GBM in this study, we have decided not to include these data in the MS. Importantly we are not claiming that the signature is specific to GBM in our MS.

Figure 1. GBM patients with high enrichment of mTOR and TAM-MG show a trend toward better survival. (A) Correlation between ssGSEA enrichment scores from mTOR signature versus TAM-MG signature in IDH-wildtype GBM (transcriptomic data from TCGA database). The survival of patients with positive mTOR and TAM-MG enrichment (orange) is compared to that of patients without this signature (green). (B) Survival of patients with high levels of RPS6 (mTOR marker) and P2RY12 (microglia-specific marker) (orange) compared to those without this signature (green). P value was calculated between the survival curve using a Log-rank test and a Wilcoxon test (displayed on the graph).

Figure 2. Correlation between ssGSEA enrichment scores from mTOR signature versus microglia or BMDM in LGG astrocytoma and oligodendroglioma. Simple linear regression analysis with the slope, r squared value and p value for the slope displayed on the graphs (transcriptomic data from TCGA database).

3. In the conditioned medium experiments (Fig. 2), it is unclear if the medium in which the different cell types (mNSCs, mGIC) are the same. Not sure if microglia media was conditioned by the tumor cells or if conditioned tumor cell media was used to grow microglia and BMDM. Are microglia media and BMDM media the same? Also, results with non-conditioned media is not presented. If these points do not come across clearly, it makes the interpretations of the results difficult for the reader.

We agree with the referee's comment that these are important points to clarify and we have now rephrased the text in both M&M (page 22) and result (page 7) sections.

In brief, mNSC and mGIC were cultured in the same medium - DMEM/F12 medium (Gibco, 31330-038), supplemented with the growth factors B27 (2%, Gibco, 12587-010), mEGF (20ng/ml, Preprotech, 315-09) and hFGF (20ng/ml, PreproTech, AF-100-18B). Culture plates were coated with poly-L-lysine (Sigma, P1524; 0.01mg/ml for 30min at room temperature-RT) and laminin (Sigma, L2020; 1:160 for 30min at 37°C). The microglia and BMDM were cultured in the same medium - DMEM media supplemented with 1% penicillin streptomycin solution. This information has now been added to the method section "Mouse cell culture".

In the *in vitro* model described in Fig.2 and Fig.S2 of the MS, the microglia and BMDM were cultured in their normal culture media (unconditioned control), or with the conditioned media of mNSC (mNSC-CM) and mGIC (mGIC-CM). No co-cultures were performed. Therefore, the experiment set up included four conditions – immune cells unconditioned, conditioned with mNSC-CM, conditioned with mGIC-CM, and conditioned with mGIC-CM plus an inhibitor. In the WB, the first lane represents the unconditioned samples and in the kinetic FACS experiments the data is normalised to the unconditioned samples (time point 0). Importantly, the differences observed in mTOR pathway activation are observed also between immune cells treated with mNSC-CM and mGIC-CM where the media is the same. This has been made clearer in the text in the results section “Glioblastoma initiating cells (GIC) increase mTOR signalling via PI3K/AKT axis in tumour-conditioned microglia but not BMDM” and in the method section “Analysis of signal transduction events”.

4. *Fig. 2: It would be great to see WB with BMDM similarly to what is shown with microglia.*

These experiments have now been performed and all results added to Fig.2/S2 and Fig.6/S6 (pages 8 and 13).

5. *It would be important to demonstrate that the key findings of figure 2 are repeated in the second mouse model that the authors used for the data shown in figure 1 and 3 for example.*

We agree this would be a valuable addition to our story and we have now performed these experiments on the GL261 model, which is the main model used in this MS. These results confirm the previous findings and can be found in Fig.2/S2 and Fig.6/S6 (pages 7 and 13).

6. *Fig 3: Absence of survival data reduces the impact about the results presented. The impact of these findings will be definitely strengthened with survival experiments demonstrating directly that disrupting mTOR pathway in microglia improves disease outcome.*

We agree with the reviewer that this is an important experiment, which would reinforce the importance of this pathway in driving tumour-promoting function of TAM-MG. We have therefore generated new cohorts of mice and followed the mice until they had to be culled because of the severity of their symptoms (survival). As shown in Fig.3D, the Cx3cr1-Rheb1^{Δ/Δ} GL261 mice survived significantly longer than the Rheb1^{fl/fl} GL261 mice (page 8).

7. *Fig. 3D: Please indicate on the figure which marker(s) are presented (Iba1, pS6, DAPI?), specifically that GL261 cells seem to also be expressing GFP and that Cre+ cells are also YFP+.*

We have added a legend in the figure to indicate which markers are presented. The fixation and antigen retrieval process quenches the flurochrome of the reporter genes – the GFP⁺ GL261 cells and the Cre⁺ YFP⁺ TAM-MG. Therefore, the isotype control showed no expression in 488nm (green) channel and so the channel was used to stain for p-S6. Further validation of the mouse model was also performed. We show a clear reduction of p-S6 levels in TAM-MG from the Cx3cr1-Rheb1^{Δ/Δ} GL261 mice upon analysis of the tumour by flow cytometry (Fig.S3D, page 9)

8. *Fig 5: The authors should make clear in the main text as well as on the figure, which type of cells (TAM-MG WT or floxed for Rheb1) were used for these experiments. Also, without corresponding legend, it is hard to know and interpret.*

We apologise for the oversight, and a clear corresponding legend has now been added. Fig.5 has been further updated and the cell type referred to in each graph has been clarified.

9. Fig. 6J-L: Similar to what is presented in figure 2, it would be important to test a control conditioned medium (e.g. conditioned with human NSCs or astrocytes) and show that the increase of p-S6 for example in microglia is specific to GBM cells.

We agree with the referee that a control conditioned medium would be desirable to confirm that the increase in p-S6 is specifically induced by GIC also in humans. However, the human *in vitro* model makes use of a patient-matched syngeneic system – where iPSC-derived iMGL were treated with hGIC-CM obtained from the same patient. While induced NSCs are commercially available and could serve as control, their use would defy the aim of this model – a syngeneic patient system. We have now repeated all experiments using media conditioned by matched-patient iPSC (hiPSC-CM) as a control. Therefore, in Fig.7 the levels of p-S6, pSTAT3 and p-NF- κ B are shown in iMGL unconditioned or treated with hiPSC-CM compared to hGIC-CM (page 16).

10. Fig. 6J-L: I would have liked to see some conditioned media experiments looking at expression level of these markers in human macrophages (derived from autologous iPSC, non-autologous blood derived primary macrophages or from monocytic leukemia lines that can be differentiated into macrophage-like cells for examples). If presented, these results would fully support the first statement of the discussion that GBM cells induce mTOR pathway specifically in TAM-MG and not in TAM-BMDM, both in murine and human models.

We appreciate the suggestion and we agree that being able to show the specificity of the pathway regulation in human TAM-MG would be desirable. Differentiation of iPSC to macrophages is not established in our laboratory and because this would be an entirely new protocol it would require longer than the revision time of the MS. However, we have carried out an *in silico* analysis on the transcriptomic TCGA datasets and shown that the mTOR pathways and the negative regulation of T cells is positively associated with TAM-MG and not with TAM-BMDM. These results can be found in Fig.7 and are described in more details in the last results section (page 15).

In order to fully support our statement that GBM cells induce mTOR pathway specifically in TAM-MG and not in TAM-BMDM, we performed tumour-conditioning (using the patient-derived GIC lines) on human blood-derived macrophages. The RosetteSep kit (StemCell, 15028) was used to extract monocytes from peripheral blood collected from healthy volunteers. The purified cells, which show expression of macrophage specific marker CD49d, were cultured with normal media, iPSC-CM, hGIC-CM or hGIC-CM+Torin. These different conditions revealed no stimulation of the mTOR pathway as demonstrated by the phosphorylation levels of S6 as well as the downstream transcription factors STAT3 and NF- κ B. The data is further described in the last result section (Fig.S7E) and in the supplementary M&M (page 6).

Although this experiment does not allow us to rule out a patient-specific effect, it support our conclusion that the TAM-MG specific mTOR regulation by GIC we describe in mouse models is translatable to the human disease context.

11. In the discussion, the authors suggest the role of Osteopontin in the phenomenon described in the study. One other potential candidate one can propose is lactate. Indeed, GBM cells are highly glycolytic and generate large amount of lactate as a result of their metabolic reprogramming supporting their proliferation. The effect of lactate on mTOR pathway has been described and could explain a lot of the results presented here and the difference between TAM-MG and TAM-BMDM may reflect a different threshold in lactate-dependent mTOR activation. The authors may want to acknowledge and discuss this.

Following the suggestion of the referee and a literature review on the effect of lactate on the mTOR pathway in macrophages/microglia and in GBM, we are now discussing the potential contribution of lactate to the described phenotype. Changes made to the text can be found in the fifth paragraph of the discussion (page 21).

Minor points:

1. Fig. 1A-C: It is unclear how many brains and how many sections per brain were analyzed and quantified. This should be included at least in the supplemental method section.

The supplementary method section "Immunohistochemistry and Tissue Analysis" were updated to include this information.

2. Fig. 2A: a drawing of a human brain is shown but the experiments were conducted with murine models. A drawing of a mouse brain may more appropriately illustrate the experiments.

We have now replaced the human brain for a mouse brain in the illustration of the experiments in Fig2A.

3. Fig. 5 lettering should be revised as it does not correspond to the main text and the legend. Because of this, the section "mTOR-dependent expression of inflammatory cytokines in tumour-conditioned microglia promotes an anti-inflammatory phenotype via the regulation of STAT3 and NF- κ B transcription factors" is difficult to follow.

We apologise for the confusion, which was due to a previous version of the figure being uploaded. Fig.5 (which is now labelled Fig.6) has now been updated and includes the missing graphs to match the figure legend and reference in the text.

4. Fig. 6 lettering shows some discrepancies with the main text and should be revised.

Fig.6 (now labelled Fig.7) has been updated and lettering has been checked to match reference in the text.

5. Page 16 first paragraph: The authors wrote: "Importantly, levels of p-NF- κ B were significantly increased under hGIC-CM and Torin treatment (Fig.6L)". Do they mean p-P65 as shown in the figure?

All analysis of the total or phosphorylated form of NF- κ B was carried out on the P65 subunit of the complex and this has now been clarified throughout the manuscript and figures.

Referee #2:

Review: Dumas et al.

In this article, Dumas et al. aim to put forward the idea that glioma-derived factors induce mTOR signaling in tumor-associated microglia but not macrophage. The study benefits from utilizing both in vivo and in vitro models of glioma-immune crosstalk including syngeneic human iPSC-derived microglia culture. The authors also propose that tumor cells should not be the primary target for mTOR inhibition. This study addresses a hitherto less studied role of mTOR signaling in the innate compartments of brain immunity in the context of GBM brain tumors. However, the work is, at best, mainly descriptive and lack any insightful mechanistic or technical breakthroughs. Several critical points need to be addressed in order to make the findings more convincing. The authors have touched upon many of the clinically relevant aspects of GBM to introduce the study but have failed to reconcile them with experimental rigor. Hence, this article is not recommended for publication in its current form.

Major comments

-The initial section chooses to address the spatial differences in mTOR signaling in TAM compartments in various genetic syngeneic mouse models of glioma but does not explain the reason behind intra model differences. The IHC images showcase PDGFB-driven model where there is no difference between the tumor core or edge. The zoomed in version does not provide any additional. The subsequent use of GL261 model does not systematically take off and spatial heterogeneity remains unaddressed.

This is an important point, which we have addressed as follows. Firstly, we have replaced the images of the PDGFB-driven model with those of the GL261 and PTEN/P53 models, which are the models we focussed on in the rest of the MS. Moreover, because the GL261 model displays a significant difference between tumour edge and core, while the PTEN/P53 model does not, the images ensure we better represent the results of the quantitative analysis. We agree with this referee that this intra-model difference is interesting. There are several possible explanations for this observation, including the different genetic mutations giving rise to the tumours which could influence the pattern of tumour growth, more or less infiltrative; alternatively the different tumour initiation method with the GL261 model being an allograft and the other models being obtained via genetic modification could play a role. We are now commenting on this in the text (page 6), although we also emphasise that a significantly higher mTOR activity in TAM is observed in all models when the tumour core is compared to non-tumour tissue and the consistency of this phenotype across models is what is most relevant to this study.

-The manuscript somewhat tries to contextualize a relatively dated transcriptomic classification scheme (Verhaak, 2010) with the current version but that does not add to the impact of the study. The detailed explanation, on the other hand, unnecessarily confuses the reader.

We agree these results are not necessary and rather confusing, they have now been removed from the MS and the results section modified accordingly.

In the light of differential overlap of Iba1+ and CX3Cr1+ microglial functions, the study should employ more functional markers for in vivo/ ex vivo phenotyping.

This is an important point, which we have addressed as follows:

For Fig.S3 and Fig.4 (now Fig.S3A-C, Fig.4B in the revised MS) the analysis on the tumour tissue was repeated using specific markers to better characterise the infiltrating immune populations. To differentiate between the TAM-MG and the TAM-BMDM, P2RY12 and CD49d markers were used respectively. P2RY12 has been shown by several groups to specifically identify microglia and not peripheral macrophages (Bowman, Klemm et al., 2016, Gosselin, Skola et al., 2017). CD49d marker was characterised by Bowman et al 2016 as a marker of peripheral macrophages in the glioblastoma tumours, which is not expressed by microglia. These results confirm and further extend our previous conclusions (page 9-10).

-Figure 2 data utilizes Pten^{-/-} p53^{-/-} GIC model to test the role of secreted factors. The study loses continuity from rather random choice of models for a particular set of experiments.

Similar experiments need to be performed with at least another GIC model.

It would be important to comment on the nature of secreted factors and their similarities/dissimilarities between different GIC models.

Western blots for TAM-BMDM would better drive home the key finding of the study.

These are very relevant points and we have now performed several experiments to address them. Firstly, a second model has been analysed; to maintain consistency WB of BMDM and MG conditioned with the secretome of GL261 have been carried out and results are shown in Fig.2/S2. Moreover, WB for conditioned BMDM were also performed with the *Pten^{-/-}/p53^{-/-}* mGIC-CM. For what it concerns the identity of the secreted factors, the upregulation of mTOR in tumour-conditioned microglia across GBM models suggest that these factors are likely consistent across different GIC lines. This is further validated by the GL261-CM generating comparable *in vitro* results to that of the *Pten^{-/-}/p53^{-/-}* mGIC-CM. We are now commenting on this point in the second results section (pages 7-8) and in the discussion (page 20), where the possibility that osteopontin and/or lactate (as suggested by referee 1) could be involved in this phenotype is mentioned.

Several figures lack important labels and information. Many of the supplemental figures find no mention in the manuscript. Figure 5 is missing two very important pieces of data mentioned in the manuscript.

These points have been addressed and rectified in this revised version of our MS. We do apologise for the confusion related to Fig.5, which was due to a previous version of the figure being uploaded. Fig.5 (which is now Fig.6) has now been updated and includes the missing graphs to match the figure legend and reference in the text.

-The authors fail to highlight the novelty of their iPSC MGL assay. The protocol is well established, and Figure 6 barely finds its way to utilize the model beyond standard characterization. The study does not comment on how it will specifically target TAM-MG mTOR signaling due any detailed novel cell-intrinsic mechanism. Geneset enrichment results are poorly explained.

We are happy to accommodate this suggestion. In particular, we agree with the referee that protocols to derive GIC are well established and now also the protocol we have used for reprogramming has been published, including for use on human cells (Gao, Nowak-Imialek et al., 2019, Yang, Ryan et al., 2019). We have therefore streamlined the description of the experimental set up and removed all data detailing the establishment and characterisation of GIC, fibroblast, iPSC lines. The figure now focusses on the use of the patient-specific tumour-conditioned iMGL model to validate our findings from mouse models. We demonstrate that factors secreted by patient-derived mesenchymal GIC lines condition iMGL to increase mTOR signalling which increases STAT3 signalling while inhibiting NF-κB signalling, as observed in different GBM mouse models.

To complement this analysis, we have taken advantage of the TCGA GBM RNAseq data to confirm in human context that increased mTOR activity in TAM-MG contributes to immune evasion mechanism by negatively regulating T cell effector function and proliferation/ infiltration in the tumour. Detail of this analysis, which aligns with findings in GBM mouse models, can be found in the last result section (page 15) and in Fig.7B-D. The last paragraph of the conclusion has also been expanded to provide more information on how we envisage that mTOR signalling could be specifically targeted in TAM-MG in a clinical context. Moreover, geneset enrichment results have been updated and more detailed explanation has been provided in the corresponding results (page 11 and 15) and supplementary method (page 10) section.

Minor Comments

-Numerous typos; non-standard term usage (Figure S5 B)

The revised version of the MS has been carefully proofread.

-Figure 3 D does not have any color coding for markers.

We have added a legend in the figure to indicate which markers are presented.

-Figure S3 A is confusing regarding the validation of loss of Rheb.

To avoid this confusion, probably due to the additional validation carried out in an *in vitro* setting (as previously displayed in Fig.S3A), we have removed these data and are now only including the most relevant validation of the model carried out *in vivo* by flow cytometry analysis of the tumour (Fig.S3A-C) and by immunofluorescence of tumour tissues (Fig.S3D).

-% of cell scale bars should include the gating strategy/information for each graph.

The gating strategy used was based on selection of single cells and live cells prior to marker selection, details have now been added to the supplementary method section 'Flow cytometry'. Representative FACS images have been added to demonstrate additional gating strategy where specific populations were identified using several markers and representative FACS plots (including % of cells with the gates where necessary) have been added.

Referee #3:

Dumas et al describe differences between the two main populations of Tumor Associated Macrophages (TAM): the bonafide infiltrating Macrophages (TAM-BMDM) and microglia (TAM-MG) in GBM. Combining data from *in vitro* systems including iPSC derived MG and GL261 allograft model, the authors describe mTOR pathway as an important regulator of TAM-MG but not TAM-BMDM. However, my main concern is that the mTOR downstream regulators that they propose need to be better characterized.

Major comments:

As mentioned above, my main concern is that the lack of evidence for the downstream regulators of mTOR, although STAT3 seems more convincing, there are not enough experimental evidence to conclude their statement "...concomitant reduction in expression of IL-12 mediated by the reduced phosphorylation of NF- κ B" (page 14, first paragraph). Torin treatment showing a modest increase on the phospho-NF- κ B is not enough to prove the connection of mTOR and NF- κ B in their specific context. Did they ever study the nuclear translocation of NF- κ B or any of the downstream targets of this transcription factor? Moreover, throughout the analysis of their transcriptional data there is not enough data linking mTOR and NF- κ B or at least they didn't highlight it.

We thank the referee for highlighting an important point. We have now carried out additional experiments, which complement and extend previous data and we strongly feel we are now providing compelling evidence of the link between mTOR and NF- κ B in TAM-MG.

- 1) In Fig.6A and S6C-D, we show a significant deregulation of NF- κ B in TAM-MG transcriptomic profile in the Cx3cr1-Rheb knockout allograft model. Components of the NF- κ B complex were found to be significantly deregulated in Cx3cr1-Rheb^{-/-} TAM-MG. This provides initial evidence of an mTOR-dependent regulation of the NF- κ B pathway.
- 2) The NF- κ B pathway is found to be most frequently associated with significantly deregulated pathways in the transcriptomic profile of Cx3cr1-Rheb^{-/-} compared to Rheb^{fl/fl} TAM-MG (Fig.6B). Fig.6D further illustrates a positive correlation between NF- κ B deregulation and the deregulated pathways including antigen presentation, Th1 and Th2 differentiation T cell chemotaxis, while being negatively correlated with mTOR signalling and the negative regulation of T cells. This suggests the NF- κ B pathway could be responsible for TAM-MG-driven functional effects observed in Cx3cr1-Rheb^{-/-} GL261 tumours.
- 3) We next showed using our mouse *in vitro* model that upon tumour-conditioning of microglia an mTOR-dependent inhibition of NF- κ B was observed, which affected the downstream expression of IL-12b.
- 4) To consolidate our argument, as proposed by the referee, the nuclear translocation of NF- κ B was investigated in our *in vitro* model. While there are no changes in BMDM in the nuclear translocation of NF- κ B under any treatment, tumour-conditioned microglia (with mGIC^{C^{Pten}-/-/p53}-/-CM or mGL261-CM) display an increase in nuclear translocation of NF- κ B under mTOR inhibition. These data substantiate our claim that mTOR inhibits NF- κ B in tumour-conditioned microglia. The data has been added in Fig.6F/S6I of the MS and is discussed page 13.
- 5) To further validate the mTOR-dependent regulation of NF- κ B *in vivo*, we performed FACS analysis of GL261 tumour tissue. We demonstrate that P2RY12⁺ CD49d⁻ TAM-MG have reduced p-S6 levels in the Cx3cr1-Rheb^{-/-} GL261 tumours and significantly reduced levels of p-NF- κ B (Fig.6I). On the other hand, the p-NF- κ B levels was not significantly changed in P2RY12⁻ CD49d⁺ TAM-BMDM in the Cx3cr1-Rheb^{-/-} GL261 tumours (Fig.6I).

Regarding the gene expression data, it is also striking to me that they didn't show a dysregulation of mTOR signalling pathway in the Rheb1 KO system (Figure 5). That can be simply since Figure 5 is missing at least 3 panels (both in the text and figure legend there is A to G but in the figure, we only see till E with the derived inconsistency on the legend and text).

We thank the referee for raising this point. The model achieves an inhibition of the mTOR pathway by KO Rheb1, the direct and only known activator of mTORC1. We demonstrate the successful inhibition of the pathway at post-translational level by immunolabelling of Rheb^{fl/fl} and Cx3cr1-Rheb^{-/-} GL261 tissues for Iba1 and p-S6. To confirm successful inhibition of the pathway in the different TAM populations, we ran a flow cytometry analysis using P2RY12 and CD49d surface markers. This analysis demonstrated a reduction in p-S6 levels in P2RY12⁺ TAM-MG (Fig.S3A-C).

Finally, we ran a GSEA enrichment analysis for the mTOR pathway (using the signature listed in Table. S3, already employed for other analysis in this MS) on the transcriptomic data of Rheb^{fl/fl} versus Cx3cr1-Rheb^{-/-} TAM-MG. The Rheb^{fl/fl} TAM-MG displayed a stronger enrichment (ES = 0.345) for the pathway than the Cx3cr1-Rheb^{-/-} TAM-MG. In keeping with these finding, bulk GL261 RNAseq data show a negative correlation between mTOR signalling and TAM in a differential correlation analysis (Fig.S5A).

In summary, our data provide robust evidence of dysregulation of mTOR signalling as a result of Rheb1 KO in the Cx3cr1-Rheb^{-/-} TAM-MG.

The authors base their conclusions on the transcriptional and functional differences between TAM-MG and TAM-BMDM. These differences are detected in vitro (Fig 2A) but less evident when they analyze TAM-MG and TAM-BMDM from tumours. In particular, the authors do not clearly explain the strategy followed to discriminate these cell types, and important point considering that they express similar cell-markers.

The referee considers the post-translational and functional differences in mTOR signalling detected *in vitro* in tumour-conditioned microglia as compared to BMDM convincing (Fig.2 and Fig.6). The transcriptional differences between TAM-MG and TAM-BMDM in tumours (*in vivo*) complement these data well and Fig.1 illustrates this point taking advantage of the RNA-sequencing data obtained from GL261 allografts (Bowman et al., 2016) where sophisticated lineage tracing approaches were used to discriminate between TAM-MG and TAM-BMDM prior to sorting and RNA-sequencing.

Moreover, we have now carried out a flow cytometry analysis of GL261 tumours using P2RY12 and CD49d surface markers to differentiate TAM-MG and TAM-BMDM populations respectively. Looking at levels of p-S6, p-STAT3 and p-NF-κB in these populations, we show a decrease in p-S6 levels, associated with a decrease in p-STAT3 and an increase in p-NF-κB levels in P2RY12⁺ CD49d⁻ TAM-MG in the Cx3cr1-Rheb^{-/-} GL261 tumours, but not in P2RY12⁻ CD49d⁺ TAM-BMDM (Fig.S3C, Fig.6H-I). These findings validate the differences between TAM-MG and TAM-BMDM profile *in vivo* using more stringent markers to differentiate the two TAM populations.

The authors should also consider and discuss whether the downregulation in TAM markers detected simply reflects decreased TAM numbers in tumours?

This is an interesting point, which we have now incorporated in the text (in the result section 'Genetic inhibition of mTOR signalling in TAM affects the immune composition of GL261 tumours').

CSF1R, Iba1 and CD11b are markers expressed by TAM-MG and TAM-BMDM and we agree with the referee that the change in their expression could theoretically be caused by a change in the number of TAM in the tumours. However, we would like to highlight that the number of TAM is differently affected with TAM-MG being less numerous and TAM-BMDM more numerous in the Cx3cr1-Rheb1 Δ/Δ tumours. Therefore, the changes in expression of the marker could also reflect a change in activity levels, in agreement with previously published observations (Bowman et al., 2016, Holtman, Raj et al., 2015).

It is important to validate the quality of iPSC-derived microglia, which in similar setups has been shown to be closer to macrophages than microglia.

The protocol used for the differentiation of the iPSC to microglia has been extensively characterised in the original publication (Muffat, Li et al., 2016). The authors demonstrated through RNAseq analysis the expression of key markers which are uniquely expressed by microglia precursors (early yolk sac myelogenesis markers) – CD41, c-Kit and CD235a – and by mature microglia – TMEM119, P2RY12, and TREM2. In this manuscript, we confirm the expression of these markers by flow cytometry and IF in the yolk sac embryoid bodies and in the iMGL, respectively (Fig.7F). Moreover, a publication by Kierdorf et al demonstrated that microglia development is dependent on Pu.1 expression and we show expression of this transcription marker in the precursors maturing into microglia (Kierdorf, Erny et al., 2013).

To further demonstrate that the differentiated cells are microglia and not monocyte-derived macrophages, we checked the expression of CD49d (a marker demonstrated to be macrophage specific and not express by microglia). The iPSC-derived iMGL did not express CD49d (data can be found in Fig.7F, Fig.S7C)

Additional comments:

Figs 2 C E are missing the data with the corresponding inhibitors. Also, as a general comment for Fig 2 C E G and I, it would be more clear if they state the inhibitor in the legend for each panel instead of making that general legend on the bottom.

As requested, the legend was added to each graph and corresponding inhibitors were added to fig.2C and E

Have the authors treated Cx3cr1:CreERT2 Rheb1 fl/fl with tamoxifen and waited a month before inducing GBM? This is an important point to assure that target genes are only deleted in MG and not macrophages.

The animals were injected with tamoxifen and a month later tumour was initiated by intracranial injection of tumour cells, as previously described in the literature (Bowman et al., 2016, Goldmann, Wieghofer et al., 2016) and this has now been clarified in the supplementary methods section 'Mice'. Moreover, as previously described (Bowman et al 2016, Goldman et al 2013), this model will predominantly target microglia with a small proportion of BMDM. We provide evidence of this with the expression of the YFP reporter gene in all TAM-MG and only in a small proportion of TAM-BMDM (Fig.S3). As discussed in the MS, although a contribution of the small proportion of TAM-BMDM targeted by KO to the observed phenotype cannot be entirely ruled out, it is thought to be unlikely as both *in vitro* and *in vivo* analysis has demonstrated lack of significant mTOR activity in TAM-BMDM.

Tables were not labeled so which table was which could be tracked.

The tables have been labelled accordingly to make it easier to track (list of table number and name can be found in the supplementary information).

References

- Bowman RL, Klemm F, Akkari L, Pyonteck SM, Sevenich L, Quail DF, Dhara S, Simpson K, Gardner EE, Iacobuzio-Donahue CA, Brennan CW, Tabar V, Gutin PH, Joyce JA (2016) Macrophage Ontogeny Underlies Differences in Tumor-Specific Education in Brain Malignancies. *Cell Rep* 17: 2445-2459
- Gao X, Nowak-Imialek M, Chen X, Chen D, Herrmann D, Ruan D, Chen ACH, Eckersley-Maslin MA, Ahmad S, Lee YL, Kobayashi T, Ryan D, Zhong J, Zhu J, Wu J, Lan G, Petkov S, Yang J, Antunes L, Campos LS et al. (2019) Establishment of porcine and human expanded potential stem cells. *Nat Cell Biol* 21: 687-699
- Goldmann T, Wieghofer P, Jordao MJ, Prutek F, Hagemeyer N, Frenzel K, Amann L, Staszewski O, Kierdorf K, Krueger M, Locatelli G, Hochgerner H, Zeiser R, Epelman S, Geissmann F, Priller J, Rossi FM, Bechmann I, Kerschensteiner M, Linnarsson S et al. (2016) Origin, fate and dynamics of macrophages at central nervous system interfaces. *Nat Immunol* 17: 797-805
- Gosselin D, Skola D, Coufal NG, Holtman IR, Schlachetzki JCM, Sajti E, Jaeger BN, O'Connor C, Fitzpatrick C, Pasillas MP, Pena M, Adair A, Gonda DD, Levy ML, Ransohoff RM, Gage FH, Glass CK (2017) An environment-dependent transcriptional network specifies human microglia identity. *Science* 356
- Holtman IR, Raj DD, Miller JA, Schaafsma W, Yin Z, Brouwer N, Wes PD, Moller T, Orre M, Kamphuis W, Hol EM, Boddeke EW, Eggen BJ (2015) Induction of a common microglia gene expression signature by aging and neurodegenerative conditions: a co-expression meta-analysis. *Acta Neuropathol Commun* 3: 31
- Kierdorf K, Erny D, Goldmann T, Sander V, Schulz C, Perdiguero EG, Wieghofer P, Heinrich A, Riemke P, Holscher C, Muller DN, Luckow B, Brouwer T, Debowski K, Fritz G, Opdenakker G, Diefenbach A, Biber K, Heikenwalder M, Geissmann F et al. (2013) Microglia emerge from erythromyeloid precursors via Pu.1- and Irf8-dependent pathways. *Nat Neurosci* 16: 273-80
- Muffat J, Li Y, Yuan B, Mitalipova M, Omer A, Corcoran S, Bakiasi G, Tsai LH, Aubourg P, Ransohoff RM, Jaenisch R (2016) Efficient derivation of microglia-like cells from human pluripotent stem cells. *Nat Med* 22: 1358-1367
- Yang J, Ryan DJ, Lan G, Zou X, Liu P (2019) In vitro establishment of expanded-potential stem cells from mouse pre-implantation embryos or embryonic stem cells. *Nat Protoc* 14: 350-378

Dear Silvia,

Thank you for submitting your revised manuscript for consideration by The EMBO Journal. Your amended study was sent back to the referees for re-evaluation. Please note that while referee #2 was at this time not able to reevaluate the revised work, we have assessed your response editorially and found his-her criticism to be reasonably considered. We have received comments on the revised manuscript from the other two reviewers, which I enclose below. As you will see they find that their concerns have been sufficiently addressed and are now broadly in favor of publication.

Thus, we are pleased to inform you that your manuscript has been accepted in principle for publication in The EMBO Journal; pending referee #1's minor remaining issue is conclusively addressed.

We also need you to consider a number of minor formatting points as listed below, which need to be adjusted at re-submission.

Please contact me at any time if you need any help or have further questions.

As you may have noticed, every paper now includes a 'Synopsis', displayed on the html and freely accessible to all readers. The synopsis includes a 'model' figure as well as 2-5 one-short-sentence bullet points that summarize the article. I would appreciate if you could provide the bullet points.

Thank you for giving us the chance to consider your manuscript for The EMBO Journal. I look forward to your final revision.

Again, please contact me at any time if you need any help or have further questions.

Kind regards,

Daniel

Daniel Klimmeck PhD
Editor
The EMBO Journal.

>> Dataset EV legends: EV tables and datasets should all have their legends directly in the files, thus they need to be removed from the appendix; Nomenclature and file needs to be updated; pls see notes below.

>> Please add all supplementary materials and methods to the main manuscript and update the

author checklist accordingly.

>> There are 8 supplementary figures. Five could be made EV figures, and the remaining three added with their legends to the appendix file. EV figure legends should be added to the main manuscript, after main figure legends.

>> Appendix File with ToC: All supplementary figures should be added to appendix file, which is to be saved as a PDF with a ToC. Please change the nomenclature to 'Appendix Figure S1, S2...' and adjust references in the main text and legends.

>> Please remove hyperlinks from the data availability section.

>> Tables S1,2 should be made datasets ("Dataset EV1" and "Dataset EV2") as they are quite large. The remaining tables would thus need to be changed to "Table EV1" - "Table EV4"

Further information is available in our Guide For Authors:

The revision must be submitted online within 90 days; please click on the link below to submit the revision online before 21st Jul 2020.

Link Not Available

Referee #1:

In their revised manuscript Dumas and colleagues have made great efforts addressing the concerns pointed out in the first round of review. This revised version of the manuscript addressed appropriately my concerns. No specific major concerns have been identified however one minor point needs to be addressed:

Point #7 of the initial round of review. Fig3D (now Fig. S2D): Image(s) of the isotype control should be presented. Additionally, the information provided in the rebuttal about the quenching of the fluorescence related to GFP and YFP in response to fixation and antigen retrieval processes, making the 488nm (green) channel available for staining, should be included in the manuscript somewhere (e.g. supplementary methods).

Referee #3:

No further comments.

Response to reviewers' comments**Referee #1:**

In their revised manuscript Dumas and colleagues have made great efforts addressing the concerns pointed out in the first round of review. This revised version of the manuscript addressed appropriately my concerns. No specific major concerns have been identified however one minor point needs to be addressed:

Point #7 of the initial round of review. Fig3D (now Fig. S2D): Image(s) of the isotype control should be presented. Additionally, the information provided in the rebuttal about the quenching of the fluorescence related to GFP and YFP in response to fixation and antigen retrieval processes, making the 488nm (green) channel available for staining, should be included in the manuscript somewhere (e.g. supplementary methods).

We have now included the isotype control image as Figure EV2E. We have also added the information regarding quenching of the fluorescence following fixation and antigen retrieval processes, as detailed in our previous response to reviewers' comments to the supplementary method (Appendix page 5 "Immunohistochemistry and tissue analysis").

Referee #3:

No further comments.

Dear Dr Marino,

Thank you for submitting the revised version of your manuscript. I have now evaluated your amended manuscript and concluded that the remaining minor concerns have been sufficiently addressed.

Thus, I am pleased to inform you that your manuscript has been accepted for publication in the EMBO Journal.

Please note that it is EMBO Journal policy for the transcript of the editorial process (containing referee reports and your response letter) to be published as an online supplement to each paper. I would like to ask for your consent on keeping the additional referee figures included in this file.

If you do NOT want this, you will need to inform the Editorial Office via email immediately. More information is available here: http://emboj.embopress.org/about#Transparent_Process

Your manuscript will be processed for publication in the journal by EMBO Press. Manuscripts in the PDF and electronic editions of The EMBO Journal will be copy edited, and you will be provided with page proofs prior to publication. Please note that supplementary information is not included in the proofs.

Should you be planning a Press Release on your article, please get in contact with embojournal@wiley.com as early as possible, in order to coordinate publication and release dates.

On a different note, I would like to alert you that EMBO Press is currently developing a new format for a video-synopsis of work published with us, which essentially is a short, author-generated film explaining the core findings in hand drawings, and, as we believe, can be very useful to increase visibility of the work.

Please see the following link for a representative example:
http://embopress.org/video_EMBOJ-2014-90147

If you have any questions, please do not hesitate to call or email the Editorial Office.

Kind regards,

Daniel Klimmeck

Daniel Klimmeck, PhD
Editor
The EMBO Journal

Corresponding Author Name: Marino

Journal Submitted to: EMBO J

Manuscript Number: EMBOJ-2019-103790R